# MMR: A Large-scale Benchmark Dataset for Multi-target and Multi-granularity Reasoning Segmentation

**Donggon Jang**[*] **Yucheol Cho**[*] **Suin Lee Taehyeon Kim Dae-Shik Kim**[†]
Department of Electrical Engineering, KAIST
{jdg900,yc_cho,suinlee,rlaxogus0814,daeshik}@kaist.ac.kr

## Abstract

The fusion of Large Language Models (LLMs) with vision models is pioneering new possibilities in user-interactive vision-language tasks. A notable application is reasoning segmentation, where models generate pixel-level segmentation masks by comprehending implicit meanings in human instructions. However, seamless human-AI interaction demands more than just object-level recognition; it requires understanding both objects and the functions of their detailed parts, particularly in multi-target scenarios. For example, when instructing a robot to *"turn on the TV"*, there could be various ways to accomplish this command. Recognizing multiple objects capable of turning on the TV, such as the TV itself or a remote control (multi-target), provides more flexible options and aids in finding the optimized scenario. Furthermore, understanding specific parts of these objects, like the TV's button or the remote's button (part-level), is important for completing the action. Unfortunately, current reasoning segmentation datasets predominantly focus on a single target object-level reasoning, which limits the detailed recognition of an object's parts in multi-target contexts. To address this gap, we construct a large-scale dataset called Multi-target and Multi-granularity Reasoning (MMR). MMR comprises 194K complex and implicit instructions that consider multi-target, object-level, and part-level aspects, based on pre-existing image-mask sets. This dataset supports diverse and context-aware interactions by hierarchically providing object and part information. Moreover, we propose a straightforward yet effective framework for multi-target, object-level, and part-level reasoning segmentation. Experimental results on MMR show that the proposed method can reason effectively in multi-target and multi-granularity scenarios, while the existing reasoning segmentation model still has room for improvement. The dataset is available at https://github.com/jdg900/MMR.

## 1 Introduction

Human-machine interaction is a key focus in AI for real-world applications, driving interest in multi-modal perception models that integrate vision and language modalities. The model perceives the context within the image related to explicit text query inputs and predicts pixel-level masks or bounding boxes accordingly. For example, Open Vocabulary Segmentation (OVS) (Liang et al., 2023; Cho et al., 2023; Xu et al., 2023), leveraging models like CLIP (Radford et al., 2021), generates segmentation masks from open-set text categories. Similarly, Referring Expression Segmentation (RES) (Wang et al., 2023; Hu et al., 2023; Liu et al., 2023a; Yang et al., 2022) predicts the segmentation mask corresponding to the objects referenced by the text input within the image. However, these models encounter challenges with implicit and complex text queries, limiting their effectiveness in real-world scenarios.

The emergence of Large Language Models (LLMs) (Zheng et al., 2024; Roumeliotis & Tselikas, 2023; Achiam et al., 2023; Zhang et al., 2023a) offers promising solutions to this challenge. Recent

---

[*]Equal Contribution
[†]Corresponding Author

studies (Bai et al., 2023; Li et al., 2023; Liu et al., 2024; Zhu et al., 2023; Zhang et al., 2023b; Chen et al., 2023; You et al., 2023) have witnessed that multimodal LLMs with superior reasoning capabilities can effectively perform vision tasks when given implicit text inputs. However, current multimodal LLMs primarily provide information corresponding to images or regions in text form, lacking pixel-level mask generation.

To address these limitations, LISA (Lai et al., 2023) introduces reasoning segmentation. Unlike previous tasks that rely on explicit text (e.g., "steak"), reasoning segmentation handles implicit queries that require intricate reasoning or world knowledge (e.g., "the food with most protein"), by combining LLMs with the Segment Anything Model (SAM) (Kirillov et al., 2023) that has robust mask generation capabilities. LISA also introduces ReasonSeg, a benchmark dataset for reasoning segmentation. ReasonSeg consists of 1,218 image-instruction pairs, each containing implicit text question-answer pairs that involve complex reasoning for each image. Nevertheless, Reason-Seg has two limitations: 1) It does not adequately address scenarios involving multiple targets, and 2) it primarily focuses on object-level reasoning, treating part-level targets ambiguously. Although the recently proposed MUSE dataset by PixelLM (Ren et al., 2023) addresses multi-target object-level reasoning, it does not consider part-level reasoning. These observations underscore that current datasets for reasoning segmentation overlook the complexities of multiple targets and part-level scenarios, concentrating instead solely on object-level reasoning. This limitation restricts more advanced functionalities in reasoning segmentation.

In this paper, we introduce a Multi-target and Multi-granularity Reasoning segmentation (MMR) dataset to overcome these limitations, which covers both multiple targets and fine-grained part-level reasoning. We collect image and mask annotations from the publicly available PACO-LVIS dataset (Ramanathan et al., 2023). These annotations include class names and bounding box information of objects and parts. Then, inspired by LLaVA (Liu et al., 2024), we generate intricate question-answer pairs using the GPT-4V API (Achiam et al., 2023). Through this, the MMR dataset contains a vast collection of 194K complex and implicit instructions for comprehensive reasoning segmentation. A distinguishing characteristic of the proposed MMR dataset is its ability to handle multiple objects and diverse parts in the question-answer pairs. This diverse granularity enables models to reason and comprehend complex questions about both multiple target objects and their parts within a single query, providing more meaningful and high-quality masks.

Moreover, we propose a simple yet effective model, Multi-target and Multi-granularity Segmentation Assistant ($M^2SA$), for multi-target, object-level, and part-level reasoning segmentation. The $M^2SA$ model incorporates an early local feature fusion and multiple [SEG] tokens, which enables the model to enhance fine-grained visual understanding and consider multi-target segmentation. Experimental results on benchmarks, such as MMR, single-target referring expression segmentation datasets, and a multi-granularity referring expression segmentation dataset, demonstrate that $M^2SA$ outperforms existing state-of-the-art methods. We believe that our dataset and model serve as a valuable resource for potential applications in real-world reasoning segmentation tasks, offering enhanced versatility and robustness.

Our contributions are summarized as follows:

- We construct the MMR dataset, which includes 194K complex and implicit question pairs for multi-target, object-level, and part-level reasoning segmentation. This dataset facilitates advanced reasoning segmentation tasks in open-world scenarios.
- We propose $M^2SA$ for multi-target, object-level, and part-level reasoning segmentation. It incorporates an early local feature fusion and multiple [SEG] tokens to improve fine-grained visual understanding and segment multiple targets.
- Experimental results on MMR and other benchmarks show that $M^2SA$ outperforms state-of-the-art methods, validating the effectiveness of its components.

## 2    RELATED WORK

**Multimodal Large Language Models**    Recent advancements (Peng et al., 2023; Taori et al., 2023; Touvron et al., 2023; Zhang et al., 2022) in multimodal Large Language Models (LLMs) have greatly improved the integration between language models and vision tasks by comprehensively understanding and recognizing multiple modalities. Recently proposed models such as BLIP-2 (Li et al., 2023),

Flamingo (Alayrac et al., 2022), MiniGPT-4 (Zhu et al., 2023), llama-adapter (Gao et al., 2023; Zhang et al., 2023a), LLaVA (Liu et al., 2024), InstructBLIP (Dai et al., 2024), InternGPT (Liu et al., 2023b), and QwenVL (Bai et al., 2023) have shown superiority at multimodal tasks such as visual question-answering and captioning, leveraging the multimodal understanding capability of LLMs. While these methods have demonstrated improved performance in vision-language tasks through instructional tuning, they only provide the text output about the visual target and focus on a holistic understanding of global information in the image. Therefore, their applicability is limited in tasks requiring finer-grained understanding at the pixel level.

**Reasoning Segmentation** The task of reasoning segmentation, introduced by LISA (Lai et al., 2023), is understanding implicit text instruction and providing a corresponding mask for the answer. This task is more challenging and important than the referring expression segmentation task which deals with explicit and simple text queries. For instance, when a user wants to segment a pepper in an image, handling an implicit query like 'the food with a spicy taste' instead of a direct reference such as 'the pepper' is significant for improving human-AI interaction. To tackle this, LISA introduces ReasonSeg, a benchmark containing implicit text queries that require complex reasoning for each image. Recently, PixelLM (Ren et al., 2023), has addressed the limitation of ReasonSeg which considers only a single target in a query text. PixelLM constructs MUSE, a new dataset with multiple target objects in the text instructions. However, both studies are still limited to object-level reasoning segmentation. Methods such as GSVA (Xia et al., 2024) and GLaMM (Rasheed et al., 2024) have also been proposed, but they focus on frameworks for object-level reasoning segmentation rather than introducing new datasets. In this paper, we extend these existing tasks and propose a new benchmark dataset that considers both part-level and object-level reasoning.

**Part-level Segmentation** Recent research (Li et al., 2022; Kirillov et al., 2019; Michieli et al., 2020; Zhou et al., 2021; Pan et al., 2023) has delved into a fine-grained understanding of objects at the part-level. For the part-level visual understanding, datasets with detailed annotations for each part are required. To this end, some initial studies (Gong et al., 2017; Li et al., 2017; Yang et al., 2019; Wah et al., 2011; Jia et al., 2020; Zheng et al., 2018) have introduced datasets with part-level masks on specific domains, such as human body parts (Gong et al., 2017; Li et al., 2017; Yang et al., 2019), bird parts (Wah et al., 2011), and fashion cloth parts (Jia et al., 2020; Zheng et al., 2018). Moreover, recognizing the need for annotations on general objects, some approaches (Chen et al., 2014; Mo et al., 2019; He et al., 2022; Zhou et al., 2019; Meletis et al., 2020; Ramanathan et al., 2023; Wei et al., 2024) have extended the existing object-level datasets by including more fine-grained annotations. Furthermore, there has been an attempt (Wang et al., 2023) to extend the previous Referring Expression Segmentation (RES) task to provide part-level segmentation masks matching explicit text queries. In line with this effort, our work introduces a new dataset that includes multiple target parts and diverse implicit text queries for multi-granularity reasoning segmentation.

## 3 MMR DATASET

Current publicly available datasets for reasoning segmentation primarily emphasize object-level reasoning. Consequently, Multimodal Large Language Models (MLLMs) often struggle with questions that involve multiple targets or require reasoning at both the object- and part-levels. To address these limitations, we introduce the Multi-target and Multi-granularity Reasoning (MMR) dataset. MMR includes multi-target, object-level, and part-level reasoning scenarios. This dataset comprises images and masks from the publicly available PACO dataset (Ramanathan et al., 2023), supplemented with implicit and complex question-answer pairs generated by the GPT-API (Achiam et al., 2023). Unlike existing datasets, MMR includes large-scale question-answer pairs that consider multiple target cases and require reasoning at both the object- and part-levels, enhancing its versatility and applicability. In the following sections, we detail the dataset generation process (Sec. 3.1), describe the data filtering process (Sec. 3.2), provide a statistical analysis of MMR (Sec. 3.3), and highlight its distinctiveness compared to existing datasets (Sec. 3.4).

### 3.1 DATA GENERATION

To generate a multi-target, object-level, and part-level reasoning segmentation dataset, we leverage the PACO-LVIS dataset (Ramanathan et al., 2023). PACO-LVIS includes 456 object-specific part

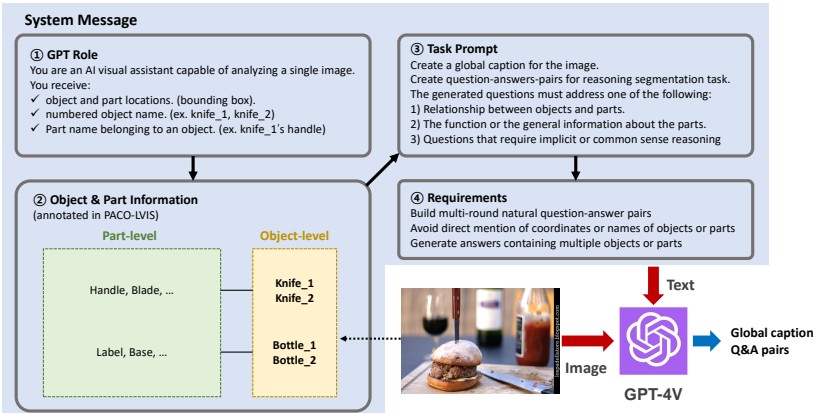

Figure 1: The prompt used in our data creation process with GPT-4V.

classes across 75 object categories, offering 502K part-level masks and bounding boxes annotated across 273K object-level masks and bounding boxes. By utilizing these comprehensive images and multi-granularity mask annotations, we can reduce annotation costs while ensuring detailed and accurate segmentation data. To create intricate and implicit question-answer pairs for multiple target and multi-granularity reasoning, we employ a GPT-assisted data generation scheme similar to LLaVA (Liu et al., 2024). Specifically, we adopt GPT-4V API which has robust visual understanding capabilities. Fig. 1 illustrates the entire data generation process.

To guide the GPT-4V API effectively, we carefully craft prompts that include ***GPT role, object and part information, task prompts, and requirements***. ***GPT role*** defines the persona of the GPT-4V API, informing it about the context and objectives of the data generation process. ***Object & part information*** provides comprehensive annotations, such as object and part names within the image and their corresponding bounding box coordinates. ***Task prompt*** informs the GPT-4V API about the task definition and considerations for generating question-answer pairs. ***Requirements*** set the rules and patterns that the GPT-4V API should follow when generating question-answer pairs (e.g., "questions should avoid direct mention of coordinates of objects or parts" or "Q&A pairs should contain multiple objects or parts"). Please see the Appendix A.5 for the detailed prompt.

The GPT-4V-assisted data generation follows a two-step process: **1) Global Caption Generation:** GPT-4V API first generates a global caption based on the image to foster a deep understanding of its context. **2) Question-Answer Pair Generation:** Leveraging this global caption along with object and part information, GPT-4V autonomously crafts multi-target, multi-granularity question-answer pairs. Carefully designed prompts and a two-step generation process enable GPT-4V to deeply comprehend image context and generate contextually relevant question-answer pairs.

## 3.2 DATA FILTERING

Despite meticulously crafted prompts for guiding GPT-4V, occasional deviations from established rules result in the generation of subpar question-answer pairs. These deviations include questions that reveal explicit target coordinates or provide overly direct hints, as well as answers that offer irrelevant information or omit essential details. To enhance the reliability of the question-answer pairs in our dataset, a rigorous filtering process is essential. Therefore, we engage four skilled human inspectors to review the dataset according to strict criteria:

- **Logicality and Reasoning**: Questions should avoid explicit target coordinates or strong hints. Non-compliant questions and their corresponding answers are removed. For example, a question like "Which part of this animal [coordinates] uses its sense of smell?" would be excluded.

- **Coherence and Relevance**: Answers lacking essential target information or containing irrelevant details are corrected for precision and relevance. This includes cases where answers mention objects or parts not provided in the annotations.

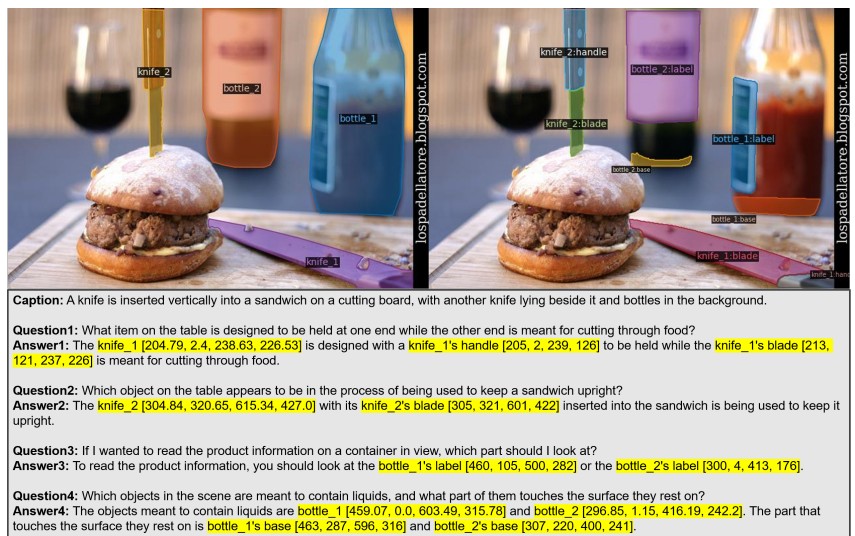

Figure 2: An example from the MMR dataset generated through our data creation process. The left and right pictures show the object- and part-level segmentation masks, respectively.

- **Clarity and Precision**: Questions and answers should be clear, concise, and free of ambiguity. For example, ill-defined data, such as asking about the function of an object or part from a segmentation perspective, is removed (e.g., "What is the function of object_1?"). Answers should provide precise information that directly addresses the question without causing confusion.

Originally, 222K question-answer pairs are generated. Of these, 12.6 % are filtered out through a review process conducted by the four inspectors, resulting in the final MMR dataset. Since dataset generation is a key contribution to our work, each inspector thoroughly reviews the entire set of 222K question-answer pairs. To minimize human error, we only filter out question-answer pairs flagged by two or more inspectors. This meticulous filtering regimen ensures the integrity and trustworthiness of the MMR dataset. An example of the generated question-answer pairs is illustrated in Fig. 2.

## 3.3 DATA STATISTICS

The MMR dataset includes 194,398 intricate and implicit question-answer pairs with 57,643 corresponding images and masks selected from PACO-LVIS. The entire dataset is split into distinct sets for training (154,127 pairs), validation (8,194 pairs), and test (32,077 pairs). Moreover, the test set is further categorized into three subsets: object-only, part-only, and mixed sets, providing a benchmark for evaluating multi-granularity reasoning segmentation capabilities.

Additionally, our dataset inherits a rich coverage of 75 object categories and 445 part categories from PACO-LVIS, enhancing its diversity and utility. We delve into the frequency distribution per object and part category across question-answer pairs. Fig. 3 (**b**) and (**d**) provide a comprehensive overview of the number of questions per object category and part category, respectively. The results show that our dataset encompasses a wide range of categories, ensuring that the question-answer pairs are not biased toward specific categories and exhibit a high level of diversity. Furthermore, the word clouds illustrated in Fig. 3 (**a**) and (**c**) highlight the prevalent head object and part categories, respectively. These word clouds demonstrate that our question-answer pairs are grounded in common and general objects and their associated parts. Fig. 3 (**e**) presents statistics on the number of targets in each question-answer pair. On average, there are 1.8 targets per answer, with the maximum number of targets in a single pair being 16. This demonstrates that our dataset can consider multiple targets in an image and cover diverse target reasoning. To evaluate the comprehensiveness of both objects and parts in the proposed dataset, we compare their occurrences within the total question-answer pairs. As depicted in Fig. 3 (**f**), there are 114,704 descriptions for objects and 226,869 for parts, maintaining a ratio of approximately 1:2. This ratio is reasonable because objects typically

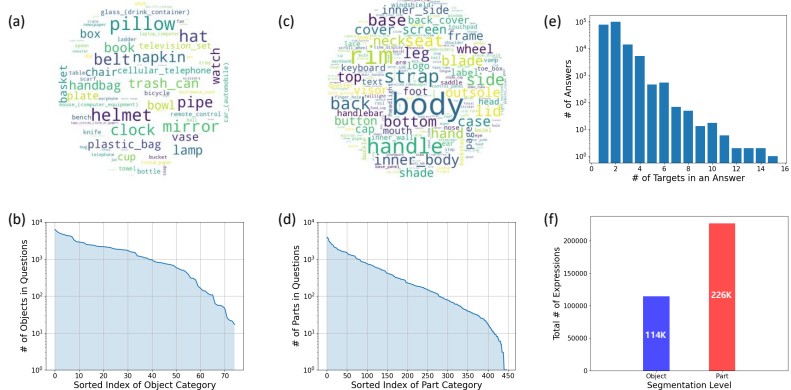

Figure 3: Statistics of the proposed MMR dataset. **(a)** the word cloud for the object categories, **(b)** the number of objects per each object category in questions (log scale), **(c)** the word cloud for the part categories, **(d)** the number of parts per each part category in questions (log scale), **(e)** the distribution of target count in answers, and **(f)** the total number of expressions of objects and parts.

Table 1: Comparison among several reasoning segmentation datasets, including ReasonSeg (Lai et al., 2023), MUSE (Ren et al., 2023), and the proposed **MMR**. Here, part-level is an expression that refers to various parts of an object that appear in the image.

| Datasets | Object-level | Part-level | Multi-target | # of Q&A pairs | GPT models |
|---|---|---|---|---|---|
| ReasonSeg | ✓ | ✓ | ✗ | 1.2K | GPT-3.5 |
| MUSE | ✓ | ✗ | ✓ | 214K | GPT-4V |
| **MMR** | ✓ | ✓ | ✓ | 194K | GPT-4V |

consist of multiple parts. Therefore, it reflects a balanced distribution, contributing to the dataset's comprehensiveness and facilitating multi-granularity knowledge understanding.

## 3.4 COMPARISON WITH EXISTING REASONING SEGMENTATION DATASETS

Tab. 1 presents a comparative overview of existing reasoning segmentation datasets and the proposed MMR dataset. As observed, MMR offers several notable advantages over existing datasets.

**First**, MMR contains 194K question-answer pairs, comparable to MUSE (Ren et al., 2023), and far exceeds ReasonSeg (Lai et al., 2023) which has only 1,218 question-answer pairs primarily designed for validation and testing purposes. This extensive scale facilitates both training and evaluation for reasoning segmentation.

**Second**, MMR supports question-answer pairs covering multi-target and multi-granularity (object-level and part-level) visual reasoning. Although MUSE includes multi-target instances, its coverage is limited to object-level reasoning. This lack of part-level detail reduces its effectiveness in fine-grained visual tasks. Part-level reasoning in MMR enables a more comprehensive understanding of visual contexts and hierarchical relationships between parts and objects. While ReasonSeg appears to include part-level reasoning, ReasonSeg often has ambiguous boundaries between objects and their parts because it doesn't specify which object a part belongs to. For instance, in a scene with a "car" and a "tire", ReasonSeg considers the "tire" as part of the "car", even if the tire is not attached. In contrast, MMR clearly distinguishes the boundaries between objects and their parts by specifying hierarchy like which object a part belongs to based on their spatial context. Additionally, unlike ReasonSeg, MMR distinguishes multiple objects of the same class within a single image at the instance level. For example, ReasonSeg might group all buses in a scene under a single "Bus" label. On the other hand, MMR treats them as distinct entities like "Bus_1," "Bus_2", etc. Also, ReasonSeg treats all screens simply as "screen," whereas MMR would specify "laptop_1's screen," "laptop_2's screen," and so forth. This allows MMR to handle objects or parts of the same class separately by considering their spatial context within the image.

**Third**, MMR leverages the advanced visual understanding capabilities of GPT-4V for question-answer generation. GPT-4V receives the image along with information such as class names and bounding boxes of objects and parts, enabling detailed and contextually accurate question-answer generation. In comparison, ReasonSeg generates questions using the language-specialized GPT-3.5 and pre-trained image tagging models, which do not fully capture the visual context, leading to less relevant question-answer pairs with the image.

**In summary**, MMR provides a substantial improvement over ReasonSeg and MUSE by including large-scale, multi-target, and multi-granularity question-answer pairs. It strengthens real-world applicability, making it a valuable asset for advancing research in reasoning-based segmentation tasks.

## 4  BASELINE FRAMEWOK

We propose a novel baseline framework for multi-target and multi-granularity reasoning segmentation, M$^2$SA. M$^2$SA enhances the LISA framework with two key features: **1) Early Local Feature Fusion** and **2) multiple [SEG] tokens.** For Early Local Feature Fusion, we extract local features from the early layer of the SAM's vision encoder, which contains fine-grained details such as image edges and boundaries. These local features are fused with the global semantic context features from the last layer of SAM's vi-

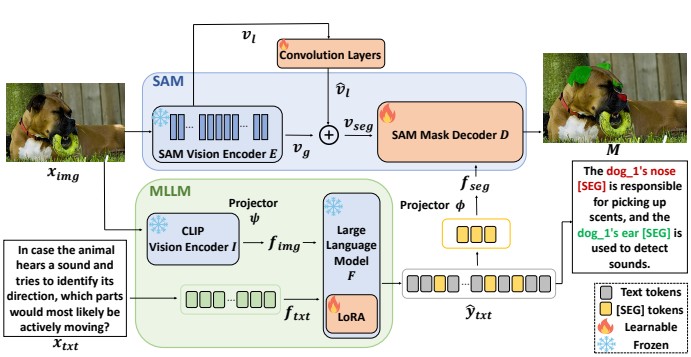

Figure 4: The overview of M$^2$SA framework.

sion encoder for more informative visual features in the mask decoder. Multiple [SEG] tokens overcome the LISA framework's limitation of a single [SEG] token, which struggles to segment multiple targets simultaneously.. To overcome this, we propose utilizing multiple [SEG] tokens. In our MMR dataset, we append a [SEG] token to each target object and part in the answer annotations (e.g., "*When closing the laptop, laptop computer's screen [SEG] would come into contact with laptop computer's base panel [SEG].*"). This approach enables the model to predict separate [SEG] tokens for each target, reducing ambiguity among multiple targets.

**Model Architecture**  Fig. 4 presents the overall architecture of the proposed M$^2$SA framework, which integrates two core components: Segment Anything Model (SAM)(Kirillov et al., 2023) and Multimodal Large Language Model (MLLM), specifically LLaVA(Liu et al., 2024). SAM module consists of SAM Vision Encoder ($E$) and SAM Mask Decoder ($D$), while the MLLM comprises CLIP Vision Encoder ($I$), vision-to-text projector ($\psi$), and Large Language Model ($F$). The image $x_{img} \in R^{h \times w \times 3}$ is fed into the SAM Vision Encoder ($E$), which generates global context features $v_g = E(x_{img}) \in R^{h/16 \times w/16 \times c}$ and early local features $v_l = E_l(x_{img}) \in R^{h/16 \times w/16 \times c'}$. To align the channel dimensions of $v_l$ with $v_g$, the early local features $v_l$ are passed through two convolution layers, resulting in refined features $\hat{v}_l \in R^{h/16 \times w/16 \times c}$. $v_g$ and $\hat{v}_l$ are then summed to obtain visual features $v_{seg} \in R^{h/16 \times w/16 \times c}$ for segmentation. Simultaneously, the image $x_{img}$ is input into the CLIP Vision Encoder ($I$), producing visual token embeddings $f_{img} = \psi(I(x_{img})) \in R^{N_{img} \times d}$, which are mapped to the LLM input space using the vision-to-text projector $\psi$. In parallel, the text queries $x_{txt}$ are tokenized by the $F$'s tokenizer, producing text token embeddings $f_{txt} \in R^{N_{txt} \times d}$. The visual token embeddings $f_{img}$ and text token embeddings $f_{txt}$ are concatenated and processed by LLM $F$, resulting in output response $\hat{y}_{txt} = F(concat(f_{img}, f_{txt}))$. $\hat{y}_{txt}$ contains the textual response to the text query and special [SEG] tokens that correspond to each target entity to be segmented. These multiple [SEG] token embeddings are extracted and projected into SAM's prompt space via the projector $\phi$, resulting in embeddings $f_{seg} = \phi(\hat{y}_{txt}[SEG]) \in R^{N_{seg} \times c}$. Finally, the SAM Mask Decoder ($D$) takes the visual features $v_{seg}$ and the multiple [SEG] token embeddings

$f_{seg}$ as input to generate the segmentation mask $M = D(concat(v_{seg}, f_{seg}))$, which identifies the target regions in the image corresponding to the text queries.

**Optimization** Our model is trained end-to-end through two sources of supervision. For the text generation, we compute auto-regressive cross-entropy loss $L_{txt}$ between the text output $\hat{y}_{txt}$ and the ground-truth text answer $y_{txt}$. For the high-quality segmentation mask generation, the mask loss $L_{mask}$ is calculated between the output mask $\hat{M}$ and the ground-truth mask $M$. The mask loss $L_{mask}$ is a weighted sum of per-pixel binary cross-entropy loss $L_{bce}$ and a DICE loss $L_{dice}$, determined by $\lambda_{bce}$ and $\lambda_{dice}$. The overall loss $L$ is formulated as follows:

$$L = L_{txt} + L_{mask},$$
$$L_{mask} = \lambda_{bce}L_{bce} + \lambda_{dice}L_{dice},$$

(1)

where $\lambda_{bce}$ and $\lambda_{dice}$ are set to 0.5 and 2.0, respectively.

## 5 EXPERIMENT

### 5.1 EXPERIMENTAL SETUP

**Implementation Details** We use pre-trained LLaVA-7B (Liu et al., 2024) and LLaVA-Llama2-13B with CLIP-ViT-L/14 (Radford et al., 2021) and Vicuna-7B (Chiang et al., 2023)/Llama2-13B (Touvron et al., 2023) to form Multimodal Large Language Model (MLLM). We adopt the pre-trained SAM-ViT-H (Kirillov et al., 2023) for the segmentation model. For CLIP-ViT-L/14, input image $x_{img}$ is resized to $224 \times 224 \times 3$ and processed with a patch size of 14, resulting in $N_{img} = 256$. LLM dimensions $d$ are set to 4096 and 5120 for Vicuna-7B and Llama2-13B. For SAM-ViT-H, $c$ and $c'$ are 256 and 1280, respectively. Efficient fine-tuning of the MLLM is facilitated using LoRA (Hu et al., 2021). The trainable components in M$^2$SA include the SAM Mask Decoder $D$, the projector $\phi$, two convolution layers, the LoRA adapter in MLLM, and the token embeddings. We use features from the 8th layer in the SAM Vision Encoder $E$ for early layer feature fusion. Our model is trained for 10 epochs, with each epoch consisting of 5,000 steps. We employ the AdamW (Loshchilov & Hutter, 2017) optimizer with a learning rate of 0.0003 and set gradient accumulation to 10 steps per update. Additionally, we use WarmupDecayLR as the learning rate scheduler. The learning rate is linearly decayed after 100 steps. The batch size and LoRA rank are set to 2 and 8, respectively. All experiments are conducted using 4 NVIDIA RTX A6000 GPUs. The results reported in the paper are the average values obtained from experiments conducted with 3 different random seeds.

**Datasets** For model training, we adopt the mixed training dataset composition scheme proposed by LISA (Lai et al., 2023), comprising four types: semantic segmentation datasets (ADE20K (Zhou et al., 2019), COCO-Stuff (Caesar et al., 2018), Mapillary (Neuhold et al., 2017), PACO-LVIS (Ramanathan et al., 2023), and PASCAL-Part (Chen et al., 2014)), referring expression segmentation datasets (RefCOCO (Kazemzadeh et al., 2014), RefCOCO+ (Kazemzadeh et al., 2014), RefCOCOg (Mao et al., 2016), and RefCLEF (Kazemzadeh et al., 2014)), a visual question answering dataset (LLaVA-Instruct-150K (Liu et al., 2024)), and the proposed MMR dataset for multi-target and multi-granularity reasoning segmentation. We sample the data from the mixed training dataset in a ratio of 2:9:2:6, where 2 represents semantic segmentation datasets, 9 represents referring expression segmentation datasets, 2 represents the visual question answering dataset, and 6 represents the proposed MMR dataset.

**Baseline Methods** To validate the effectiveness of the M$^2$SA for a multi-target and multi-granularity reasoning segmentation task, we adopt LISA (Lai et al., 2023), GSVA (Xia et al., 2024), and GLaMM (Rasheed et al., 2024) along with their variants. The pre-trained models refer to those trained solely on their respective datasets. In contrast, the variant models referred to as model$_{tr}$, are trained from scratch on a mixed training set that includes the MMR dataset. Due to issues with the publicly available code from the PixelLM, we exclude PixeLM from the baseline methods to ensure reliable and consistent comparison results. For a Multi-granularity Referring Expression Segmentation (MRES) task, we additionally adopt the class RES models (Yang et al., 2022; Liu et al., 2023a; Wang et al., 2023; 2022) and the general models (Zhu et al., 2022; Zou et al., 2023; 2024).

Table 2: Results on MMR benchmark. The gIoU and cIoU metrics are reported for the comparison. Obj & Part, Obj, and Part denote multi-granularity, object-only, and part-only evaluation settings. The best results are highlighted in **bold**.

| Methods | val | | | | test | | | |
| --- | --- | --- | --- | --- | --- | --- | --- | --- |
| | Obj & Part | | Obj | | Part | | Obj & Part | |
| | gIoU | cIoU | gIoU | cIoU | gIoU | cIoU | gIoU | cIoU |
| LISA-7B (Lai et al., 2023) | 13.8 | 18.3 | 23.5 | 25.1 | 6.6 | 7.9 | 14.5 | 17.9 |
| LISA-7B$_{tr}$ | 19.4 | 31.6 | 34.7 | 41.8 | 8.0 | 13.1 | 19.5 | 27.1 |
| GSVA-7B (Xia et al., 2024) | 14.6 | 25.1 | 26.4 | 34.3 | 6.0 | 11.6 | 15.5 | 24.8 |
| GSVA-7B$_{tr}$ | 19.8 | 38.9 | 30.2 | 41.1 | 8.0 | 18.6 | 21.2 | 34.5 |
| GLaMM (Rasheed et al., 2024) | 12.6 | 19.2 | 23.7 | 31.9 | 3.9 | 6.4 | 13.3 | 18.7 |
| GLaMM$_{tr}$ | 26.9 | 47.1 | 40.3 | 54.2 | 12.1 | 25.5 | 30.3 | 45.0 |
| **M$^2$SA-7B** | **27.8** | **48.6** | **41.0** | **55.6** | **13.5** | **27.0** | **30.9** | **46.8** |
| LISA-Llama2-13B (Lai et al., 2023) | 15.4 | 20.0 | 26.1 | 27.9 | 7.4 | 8.4 | 16.1 | 19.8 |
| LISA-Llama2-13B$_{tr}$ | 22.3 | 33.4 | 40.2 | 45.2 | 10.7 | 16.4 | 23.0 | 29.2 |
| **M$^2$SA-Llama2-13B** | **28.4** | **49.1** | **42.3** | **57.6** | **13.6** | **27.2** | **31.6** | **47.6** |

Table 3: Referring expression segmentation results on RefCOCO, RefCOCO+ (Kazemzadeh et al., 2014) and RefCOCOg (Mao et al., 2016) among M$^2$SA and existing methods. For a fair comparison with previous methods, the cIoU metrics are adopted. The best results are highlighted in **bold**.

| Methods | RefCOCO | | | RefCOCO+ | | | RefCOCOg | |
| --- | --- | --- | --- | --- | --- | --- | --- | --- |
| | val | testA | testB | val | testA | testB | val(U) | test(U) |
| MCN (Luo et al., 2020) | 62.4 | 64.2 | 59.7 | 50.6 | 55.5 | 44.7 | 49.2 | 49.4 |
| VLT (Ding et al., 2021) | 67.5 | 70.5 | 65.2 | 56.3 | 61.0 | 50.1 | 55.0 | 57.0 |
| CRIS (Wang et al., 2022) | 70.5 | 73.2 | 66.1 | 62.3 | 68.1 | 53.7 | 59.9 | 60.4 |
| LAVT (Yang et al., 2022) | 72.7 | 75.8 | 68.8 | 62.1 | 68.4 | 55.1 | 61.2 | 62.1 |
| ReLA (Liu et al., 2023a) | 73.8 | 76.5 | 70.2 | 66.0 | 71.0 | 57.7 | 65.0 | 66.0 |
| X-Decoder (Zou et al., 2023) | - | - | - | - | - | - | 64.6 | - |
| SEEM (Zou et al., 2024) | - | - | - | - | - | - | 65.7 | - |
| LISA-7B (Lai et al., 2023) | 74.1 | 76.5 | 71.1 | 62.4 | 67.4 | 56.5 | 66.4 | 68.5 |
| GSVA-7B (Xia et al., 2024) | 76.4 | 77.4 | 72.8 | 64.5 | 67.7 | 58.6 | 71.1 | 72.0 |
| GLaMM (Rasheed et al., 2024) | **79.5** | **83.2** | **76.9** | **72.6** | **78.7** | **64.6** | **74.2** | **74.9** |
| **M$^2$SA-7B** | 74.0 | 76.8 | 69.7 | 63.1 | 67.2 | 56.1 | 67.0 | 68.3 |
| LISA-Llama2-13B (Lai et al., 2023) | 73.6 | 77.3 | 70.5 | 63.2 | **68.2** | 57.0 | 67.0 | 68.4 |
| **M$^2$SA-Llama2-13B** | **74.6** | **77.6** | **71.0** | **64.0** | 68.1 | **57.6** | **69.0** | **69.3** |

**Evaluation Metrics** Following the implementation of the referring expression segmentation works, we adopt gIoU and cIoU scores to assess the quality of the segmentation mask. gIoU denotes the mean IoU for each mask, whereas cIoU is computed by the cumulative intersection area over the cumulative union area across the entire dataset. Given that cIoU may exhibit bias towards large-area objects, gIoU is preferable for evaluating part regions.

## 5.2 RESULTS ON BENCHMARK DATASETS

**Comparison on MMR** Tab. 2 compares M$^2$SA and the baseline models in a multi-target and multi-granularity reasoning segmentation task (MMR dataset). The pre-trained models perform poorly on the proposed MMR dataset, particularly struggling with the part-only set due to its lack of detailed part-level understanding. Conversely, LISA$_{tr}$, GSVA$_{tr}$, and GLaMM$_{tr}$, trained using the proposed MMR dataset, exhibit superior performance as they acquire both object-level and part-level knowledge. However, its ability to handle multi-target and fine-detail reasoning remains limited. In contrast, the proposed M$^2$SA shows highly competitive performance, effectively managing multi-target scenarios and fine-detail tasks, thus showcasing its strength in comprehensive reasoning segmentation. Qualitative results are provided in the Appendix A.13.

**Comparison on Referring Expression Segmentation Task** Tab. 3 presents the single-target object-level RefCOCO series dataset results. While M$^2$SA achieves commendable performance. it is important to note that single-target referring expression segmentation is a relatively simple task, involving explicit queries that focus on identifying a single object. The true strength of M$^2$SA lies in its ability to excel in more complex and challenging tasks, such as multi-target referring expression segmentation and multi-granularity referring segmentation. To evaluate its performance on multi-target referring expression segmentation, we curate text queries for multi-target objects using annotation information from the RefCOCO-series datasets. Each query is constructed by randomly selecting 4 to 6 object categories from each image and generating text prompts like *"Can you segment the class 1, class 2, . . ., and class n?"*. We then compare M$^2$SA's performance against LISA,

Table 4: Multi-referring expression segmentation results. We adopt the cIoU metric for comparison. The best results are highlighted in **bold**.

| Methods | Multi-RefCOCO | | | Multi-RefCOCO+ | | | Multi-RefCOCOg | |
|---|---|---|---|---|---|---|---|---|
| | val | testA | testB | val | testA | testB | val(U) | test(U) |
| LISA-7B (Lai et al., 2023) | 34.0 | 32.7 | 36.4 | 28.2 | 28.6 | 28.5 | 45.2 | 48.7 |
| GSVA-7B (Xia et al., 2024) | 50.7 | 53.3 | 47.8 | 44.8 | 47.4 | 40.6 | 47.7 | 48.6 |
| GLaMM (Rasheed et al., 2024) | 30.8 | 32.0 | 30.0 | 28.8 | 29.6 | 27.2 | 32.5 | 35.0 |
| **M²SA-7B** | **71.3** | **73.3** | **67.2** | **61.8** | **65.3** | **55.8** | **62.0** | **63.6** |
| LISA-Llama2-13B (Lai et al., 2023) | 33.2 | 32.6 | 32.4 | 27.7 | 29.9 | 26.7 | 44.0 | 47.1 |
| **M²SA-Llama2-13B** | **72.0** | **75.6** | **68.0** | **62.3** | **67.1** | **56.1** | **65.4** | **65.8** |

Table 5: Multi-granularity referring expression segmentation results on RefCOCOm (Wang et al., 2023). For a fair comparison with previous methods, the mIoU metrics are adopted. Part denotes part-only evaluation, and Obj & Part denotes multi-granularity evaluation. The best results are highlighted in **bold**.

| Methods | val | | testA | | testB | |
|---|---|---|---|---|---|---|
| | Part | Obj & Part | Part | Obj & Part | Part | Obj & Part |
| SeqTR (Zhu et al., 2022) | 13.9 | 28.2 | 12.1 | 22.8 | 18.1 | 34.7 |
| CRIS (Wang et al., 2022) | 10.6 | 25.4 | 10.1 | 21.2 | 12.9 | 30.0 |
| LAVT (Yang et al., 2022) | 15.3 | 29.9 | 13.2 | 24.4 | 18.7 | 35.5 |
| X-Decoder (Zou et al., 2023) | 16.2 | 29.5 | 13.6 | 23.6 | 20.3 | 33.8 |
| SEEM (Zou et al., 2024) | 16.1 | 29.4 | 13.6 | 23.4 | 20.4 | 33.9 |
| UniRES (Wang et al., 2023) | 19.6 | 34.3 | 16.4 | 27.8 | 25.2 | **41.7** |
| LISA-7B (Lai et al., 2023) | 21.3 | 34.3 | 18.5 | 28.6 | 25.7 | 40.1 |
| GSVA-7B (Xia et al., 2024) | 11.4 | 23.1 | 9.2 | 19.2 | 16.8 | 28.2 |
| GLaMM (Rasheed et al., 2024) | 21.4 | 35.3 | 18.6 | 29.5 | 26.9 | 41.1 |
| **M²SA-7B** | **22.4** | **35.5** | **19.9** | **30.1** | **27.1** | 41.4 |
| LISA-Llama2-13B (Lai et al., 2023) | 22.1 | 35.2 | 19.4 | 29.7 | 27.2 | 41.6 |
| **M²SA-Llama2-13B** | **24.5** | **37.3** | **21.9** | **31.9** | **28.5** | **42.7** |

GSVA, and GLaMM. As shown in Tab. 4, M²SA significantly outperforms these methods, showcasing its ability to reason about multiple objects simultaneously and effectively leverage its multi [SEG] tokens for diverse and intricate queries.

Additionally, we evaluate M²SA on RefCOCOm, a multi-granularity referring segmentation dataset. As demonstrated in Tab. 5, M²SA surpasses existing methods in this task, though the performance improvement is less pronounced. This is likely because the MMR dataset does not include the *person* class, which constitutes a significant portion of the categories in RefCOCOm. These results emphasize the versatility and effectiveness of M²SA in addressing complex, real-world scenarios, extending well beyond simple single-target segmentation tasks.

## 6 CONCLUSION

This paper addresses the limitations of current reasoning segmentation datasets, which often overlook multi-target or part-level reasoning. To resolve these issues, we introduce the Multi-target and Multi-granularity Reasoning (MMR) dataset, providing 194K comprehensive question-answer pairs that cover multi-target, object-level, and part-level aspects, enhancing diverse and context-aware interactions. We also propose the M²SA model, designed for multi-target, object-level, and part-level reasoning segmentation. M²SA incorporates early local feature fusion and multiple [SEG] tokens, improving fine-grained visual understanding and multi-target segmentation. Experimental results show that M²SA outperforms existing models on the MMR benchmark. The MMR dataset aims to drive progress in reasoning segmentation by emphasizing the importance of multi-target and part-level aspects in human-AI interactions.

## ACKNOWLEDGMENTS

This research has been supported by the LG Electronics Corporation. (Project No. G01230381)

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

# A    APPENDIX

## A.1    LIMITATION

While PACO-LVIS provides diverse and comprehensive object-part mask annotations for common objects, it lacks information on the human class and its parts. Consequently, our question-answer pairs generated based on PACO-LVIS do not consider reasoning about human class and its parts, which is a drawback. Therefore, there is a need for future dataset expansion to include a wider range of objects and parts that exist in real-world environments. Additionally, although we carefully design the prompts to ensure the diversity and quality of the dataset, the content of the question-answer pairs is inherently dependent on the pre-trained knowledge of ChatGPT.

## A.2    ETHICS CONCERN

The MMR dataset is constructed based on the publicly available PACO-LVIS dataset (Ramanathan et al., 2023), which helps mitigate privacy concerns. As the objects and parts within the images are already annotated, we only add text question-answer pairs, ensuring that potential privacy issues remain minimal. These question-answer pairs are generated using the ChatGPT/GPT-4V API (Achiam et al., 2023). While there is a risk of bias from the training data of the ChatGPT/GPT-4V API, we have implemented a thorough data filtering process to remove any ethically problematic content.

## A.3    LICENSE

We utilize the released code from LISA (Lai et al., 2023) for the baseline model code construction. Since LISA follows Apache License 2.0, our code is also licensed under Apache License 2.0. Additionally, the PACO-LVIS dataset is licensed under a Creative Commons Attribution 4.0 (CC BY 4.0) license. Consequently, our MMR dataset is also licensed under Creative Commons Attribution 4.0 (CC BY 4.0). To download the PACO-LVIS dataset (Ramanathan et al., 2023), we utilize author-released code under the MIT license. We use ChatGPT/GPT-4V API (Achiam et al., 2023) developed by OpenAI to generate the question-answer pairs in the MMR dataset. Specific licensing information for the ChatGPT/GPT-4V API model is proprietary to OpenAI.

## A.4    THE SPECIFIC DETAILS OF CHATGPT API

The specific command to use the ChatGPT API (Achiam et al., 2023) for generating question-answer pairs in MMR is as follows:

```
response = openai.Completion.create
(
    model="gpt-4-vision-preview",
    messages=prompt,
    temperature=0.7,
    max_tokens=850,
    )
```

Figure 5: To generate question-answer pairs in MMR dataset, we use gpt-4-vision-preview model. For the hyper-parameters, we set the temperature to 0.7 and max_tokens to 850.

## A.5    PROMPTS AND EXAMPLES

**General MMR Dataset**    The MMR dataset fundamentally includes multi-target (both objects and parts) answers to each question. In this section, we discuss the full prompt not covered in the main manuscript. Fig. 6 illustrates the prompt used to generate the train, validation, and test datasets. Both text and image prompts are input into GPT-4V (Achiam et al., 2023), resulting in the creation of question-answer pairs that encompass various information about objects and parts. As shown in Fig. 2, the output includes a global caption and question-answer pairs for the image. The

"You are an AI visual assistant capable of analyzing a single image. You receive the specific object locations and part locations within the image, along with detailed coordinates. These coordinates are in the form of bounding boxes, represented as (x1, y1, x2, y2). These values correspond to the top left x, top left y, bottom right x, and bottom right y. The height and width of the image you receive are 427 and 640, respectively. Additionally, there may be multiple objects of the same category in the image. To resolve this ambiguity, we use "object_number" such as "person_1" and "person_2" to differentiate between objects of the same category. If a region is a part of an object, the category name is described as "object's part", like "person's body" and "bus's wheel". The category names and bounding box coordinates of objects and parts are as follow:

"""
bottle_1 [459.07, 0.0, 603.49, 315.78];
bottle_1's label [460, 105, 500, 282];
bottle_1's neck [470, 0, 593, 62];
bottle_1's shoulder [461, 56, 603, 103];
bottle_1's body [460, 94, 604, 291];
bottle_1's base [463, 287, 596, 316];
bottle_2 [296.85, 1.15, 416.19, 242.2];
bottle_2's base [307, 220, 400, 241];
bottle_2's label [300, 4, 413, 176];
bottle_2's body [307, 172, 403, 231];
knife_1 [204.79, 2.4, 238.63, 226.53];
knife_1's blade [213, 121, 237, 226];
knife_1's handle [205, 2, 239, 126];
knife_2 [304.84, 320.65, 615.34, 427.0];
knife_2's blade [305, 321, 601, 422];
knife_2's handle [529, 399, 616, 426];
"""

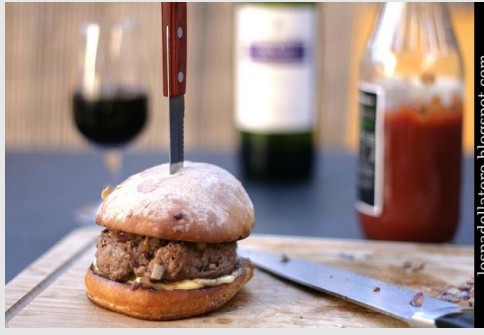

You first need to create a global caption for the image without given information. A global caption should summarize the content of the image within a maximum of two sentences. The format for the global caption strictly follows: "Global caption: GLOBAL_CAPTION_FOR_THE_IMAGE. What you need to do next is create question-answers-pairs using the information of objects and parts given above. However, when the corresponding object and part name appear in the answers, "name [coordinates]" is used as the given information above without changing its form. The goal of generating the question-answers-pair is to use the provided information about objects and object's parts, create a plausible and challenging question about the image, and provide the answer in detail for the image reasoning segmentation. The content of the question must address one of the following two:
1) the relationship between parts within the image or the relationship between a part and an object.
2) the function or the general information about the parts.

The question should be implicit and require commonsense reasoning, rather than explicitly mentioning the names of the object and part. In other words, it's important to make the question challenging by not directly including visual content details. The answer should include multiple object's parts. You must build at least 3 rounds of natural question-answer pairs, and if there is sufficient information create up to 5 rounds of question-answer pairs. In addition, please follow the format strictly: The order must be attached to the questions and answers like Question 1: and Answer 1:. In the answer, the coordinates referring to the target part or object must be attached to the object name or part name in the format: object_1 [x1, y1, x2, y2] and object_1's part [x1, y1, x2, y2]. Do not use other format such as "a part of object_1. Here are some additional requirements about generated question and answers:
1. Do not mention that the information source is provided in text description. Always answer as if you are directly looking at the image.
2. Do not ask the question you are not confident to answer. Only include question that have definite answer.
3. Do not mention the coordinates of a part and an object directly in the question.
4. Make the questions and answers concise and easy to understand, avoiding overly complex and ambiguous sentences.
5. The question should describe a complete activity, a function, or general information.
6. The answer to the generated question should include at least two object's parts and explicitly describe the names of the part and the object. Implying other potential parts is strictly prohibited.
7. Even if the image includes the real people and the brand name, or is not associated with the mentioned information, make sure to still create the question-answer pairs.
8. Avoid using incorrectly formatted object names or part names, such as located at [coordinates] or a part [object_1's part [coordinates]]. In other words, use it as it appears in the object and part information given above. ### For example: shoe_1's outsole [42, 332, 336], not an outsole [shoe_1's outsole [42, 332, 62, 336]].###
9. All generated answers must include the given object or part information, without changing the format. "

Figure 6: The text and image prompt used in our data creation for MMR dataset with GPT-4V.

segmentation mask information for the objects or parts mentioned in the answers is sourced from PACO-LVIS (Ramanathan et al., 2023) to create new annotations.

**Part-only MMR Test Dataset**  The MMR dataset includes a substantial amount of information on parts to enhance part-level recognition, which has been overlooked in existing reasoning segmentation datasets. Consequently, we create a part-level test dataset to evaluate part-level recognition separately. Using the text and image prompts shown in Fig. 7, we generate a part-only test dataset from 2000 images with extensive part-level information from PACO-LVIS annotations. As shown in Fig. 8, the output includes a global caption and question-answer pairs for the image. The segmentation mask information for the parts mentioned in the answers is sourced from the PACO-LVIS test dataset to create new annotations.

**Object-only MMR Test Dataset**  To evaluate recognition separately for object-level, we create an MMR test dataset that includes only information on objects. We generate an object-only test dataset using the text and image prompts shown in Fig. 9, selecting 2000 images with minimal part-level information. As shown in Fig. 10, the output includes a global caption and question-answer pairs for

"You are an AI visual assistant capable of analyzing a single image. You receive the specific object's part locations within the image, along with detailed coordinates. These coordinates are in the form of bounding boxes, represented as (x1, y1, x2, y2). These values correspond to the top left x, top left y, bottom right x, and bottom right y. The height and width of the image you receive are 428 and 640, respectively. Additionally, there may be multiple objects of the same category in the image. To resolve this ambiguity, we use "object_number" such as "person_1" and "person_2" to differentiate between objects of the same category. If a region is a part of an object, the category name is described as "object's part", like "person's body" and "bus's wheel". The category names and bounding box coordinates of parts are as follow:

""
dog_1's eye [235, 67, 291, 100];
dog_1's ear [324, 36, 426, 145];
dog_1's nose [184, 98, 212, 127];
dog_1's teeth [245, 146, 285, 171];
dog_1's head [169, 20, 427, 202];
dog_1's foot [337, 204, 510, 407];
dog_1's leg [212, 95, 542, 356];
dog_1's body [243, 20, 503, 328];
bowl_1's rim [143, 298, 369, 378];
bowl_1's inner_body [150, 302, 361, 371];
bowl_1's bottom [194, 362, 308, 376];
bowl_1's body [153, 351, 354, 422];

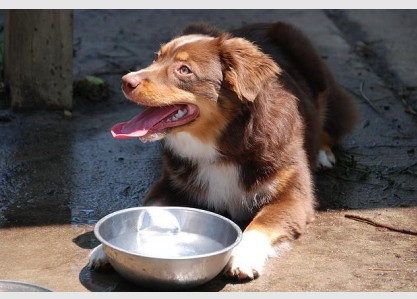

""
You first need to create a global caption for the image without given information. A global caption should summarize the content of the image within a maximum of two sentences. The format for the global caption strictly follows: "Global caption: GLOBAL_CAPTION_FOR_THE_IMAGE. What you need to do next is create question-answers-pairs using the information of object's parts given above. However, when the corresponding object's part name appear in the answers, "name [coordinates]" is used as the given information above without changing its form. The goal of generating the question-answers-pair is to use the provided information about object's parts, create a plausible and challenging question about the image, and provide the answer in detail for the image reasoning segmentation. The content of the question must address one of the following two:
1)  the relationship between different parts within the image.
2) the function or the general information about the parts.

The question should be implicit and require commonsense reasoning, rather than explicitly mentioning the names of the object and part. In other words, it's important to make the question challenging by not directly including visual content details. The answer should include multiple object's parts. You must build at least 3 rounds of natural question-answer pairs, and if there is sufficient information create up to 5 rounds of question-answer pairs. In addition, please follow the format strictly: The order must be attached to the questions and answers like Question 1: and Answer 1:. In the answer, the coordinates referring to the target part must be attached to the part name in the format: object_1's part [x1, y1, x2, y2]. Do not use other format such as "a part of object_1. Here are some additional requirements about generated question and answers:
1. Do not mention that the information source is provided in text description. Always answer as if you are directly looking at the image.
2. Do not ask the question you are not confident to answer. Only include question that have definite answer.
3. Do not mention the coordinates of a part and an object directly in the question.
4. Make the questions and answers concise and easy to understand, avoiding overly complex and ambiguous sentences.
5. The question should describe a complete activity, a function, or general information.
6. The answer to the generated question should include at least two object's parts and explicitly describe the names of the part. Implying other potential parts is strictly prohibited.
7. Even if the image includes the real people and the brand name, or is not associated with the mentioned information, make sure to still create the question-answer pairs.
8. Avoid using incorrectly formatted part names, such as located at [coordinates] or a part [object_1's part [coordinates]]. In other words, use it as it appears in the part information given above. ### For example: shoe_1's outsole [42, 332, 62, 336], not an outsole [shoe_1's outsole [42, 332, 62, 336]].###
9. All generated answers must include the given part information, without changing the format.
10. When creating questions, ask only questions about the object's parts given above without directly mentioning the part name in the question. Please keep in mind that other parts should not dominate the answer.
11. If the number of object's parts given for an image is large enough, create a question so that each round's answer includes different object's parts.
12. Do not create questions that are answered by parts other than the part information given above.
13. If that part doesn't directly answer the question, do not mention it in the answer."

Figure 7: The text and image prompt used in our data creation for the part-only MMR test dataset with GPT-4V.

the image. The segmentation mask information for the objects mentioned in the answers is sourced from the PACO-LVIS test dataset to create new annotations.

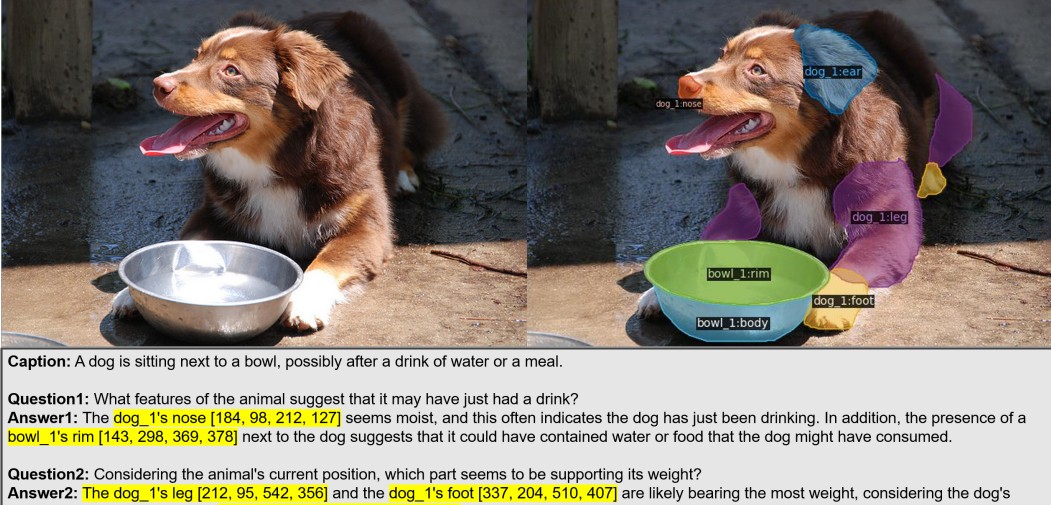

Figure 8: An example from the part-only MMR test dataset generated through the prompt in Fig. 7. This example includes information of some object's parts. The left and right pictures show the original image and part-level segmentation masks, respectively.

### A.6  DATA FORMAT

The MMR dataset is given in JSON format. The JSON file for each instance is organized as shown in Fig. 11.

Table 6: The effect of multiple [SEG] Tokens and Early Local Feature Fusion in M$^2$SA-7B on MMR benchmark. Obj & Part, Obj, and Part denote multi-granularity, object-only, and part-only evaluation settings.

| multiple [SEG] Tokens | Early Local Feature Fusion | val | | | | test | | | |
|---|---|---|---|---|---|---|---|---|---|
| | | Obj & Part | | Obj | | Part | | Obj & Part | |
| | | gIoU | cIoU | gIoU | cIoU | gIoU | cIoU | gIoU | cIoU |
| ✗ | ✗ | 19.4 | 31.6 | 34.7 | 41.8 | 8.0 | 13.1 | 19.5 | 27.1 |
| ✔ | ✗ | 26.0 | 47.7 | 39.5 | 55.4 | 11.7 | 25.2 | 28.4 | 45.2 |
| ✔ | ✔ | **27.9** | **48.5** | **41.0** | **55.6** | **13.5** | **27.0** | **31.0** | **46.8** |

### A.7  EFFECTIVENESS OF THE MULTIPLE [SEG] TOKENS AND EARLY LOCAL FEATURE FUSION

We conduct an ablation study to verify the effectiveness of the multiple [SEG] tokens and Early Local Feature Fusion proposed in M$^2$SA. Tab. 6 demonstrates that merely adding multiple [SEG] tokens results in significant performance improvements in MMR evaluation benchmarks. This finding suggests that using single [SEG] tokens in the LISA is inadequate to fully capture the segmentation capability. Moreover, performance improvements are evident when Early Local Feature Fusion is incorporated. Notably, there is a substantial performance enhancement in the part-only evaluation setting of the MMR test set. This improvement likely arises because Early Layer features contain local detail information (e.g., edges or boundaries), which aids in part and fine-level segmentation.

"You are an AI visual assistant capable of analyzing a single image. You receive the specific object locations within the image, along with detailed coordinates. These coordinates are in the form of bounding boxes, represented as (x1, y1, x2, y2). These values correspond to the top left x, top left y, bottom right x, and bottom right y. The height and width of the image you receive are 333 and 500, respectively. Additionally, there may be multiple objects of the same category in the image. To resolve this ambiguity, we use "object_number" such as "person_1" and "person_2" to differentiate between objects of the same category. The category names and bounding box coordinates of objects are as follow:

"""

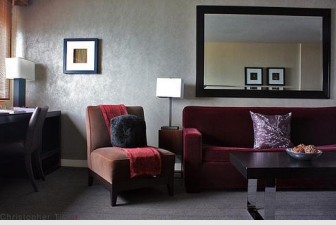

mirror_1 [304.9, 35.42, 476.09, 146.49];
pillow_1 [169.86, 180.73, 221.21, 229.7];
pillow_2 [370.81, 175.9, 436.81, 231.85];

"""

You first need to create a global caption for the image without given information. A global caption should summarize the content of the image within a maximum of two sentences. The format for the global caption strictly follows: "Global caption: GLOBAL_CAPTION_FOR_THE_IMAGE. What you need to do next is create question-answers-pairs using the information of objects given above. However, when the corresponding object name appear in the answers, "name [coordinates]" is used as the given information above without changing its form. The goal of generating the question-answers-pair is to use the provided information about objects, create a plausible and challenging question about the image, and provide the answer in detail for the image reasoning segmentation. The content of the question must address one of the following two:
1)  the relationship between objects within the image.
2) the function or the general information about the objects.

The question should be implicit and require commonsense reasoning, rather than explicitly mentioning the names of the object. In other words, it's important to make the question challenging by not directly including visual content details. Some of the answers of the rounds should include multiple different types of objects. You must build at least 3 rounds of natural question-answer pairs, and if there is sufficient information create up to 5 rounds of question-answer pairs. In addition, please follow the format strictly: The order must be attached to the questions and answers like Question 1: and Answer 1:. In the answer, the coordinates referring to the target object must be attached to the object name in the format: object_1 [x1, y1, x2, y2]. Here are some additional requirements about generated question and answers:

1. Do not mention that the information source is provided in text description. Always answer as if you are directly looking at the image.
2. Do not ask the question you are not confident to answer. Only include question that have definite answer.
3. Do not mention the coordinates of an object directly in the question.
4. Make the questions and answers concise and easy to understand, avoiding overly complex and ambiguous sentences.
5. The question should describe a complete activity, a function, or general information.
6. The answer to the generated question should include at least two objects and explicitly describe the names of the object. Implying other potential objects is strictly prohibited.
7. Even if the image includes the real people and the brand name, or is not associated with the mentioned information, make sure to still create the question-answer pairs.
8. Avoid using incorrectly formatted object names, such as located at [coordinates] or an object_1 [object_1 [coordinates]]. In other words, use it as it appears in the object information given above.
9. All generated answers must include the given object information, without changing the format.
10. When creating questions, ask only questions about the objects given above without directly mentioning the object name in the question. Please keep in mind that other objects should not dominate the answer.
11. If the number of objects given for an image is large enough, create a question so that each round's answer includes different objects."

Figure 9: The text and image prompt used in our data creation for the object-only MMR test dataset with GPT-4V.

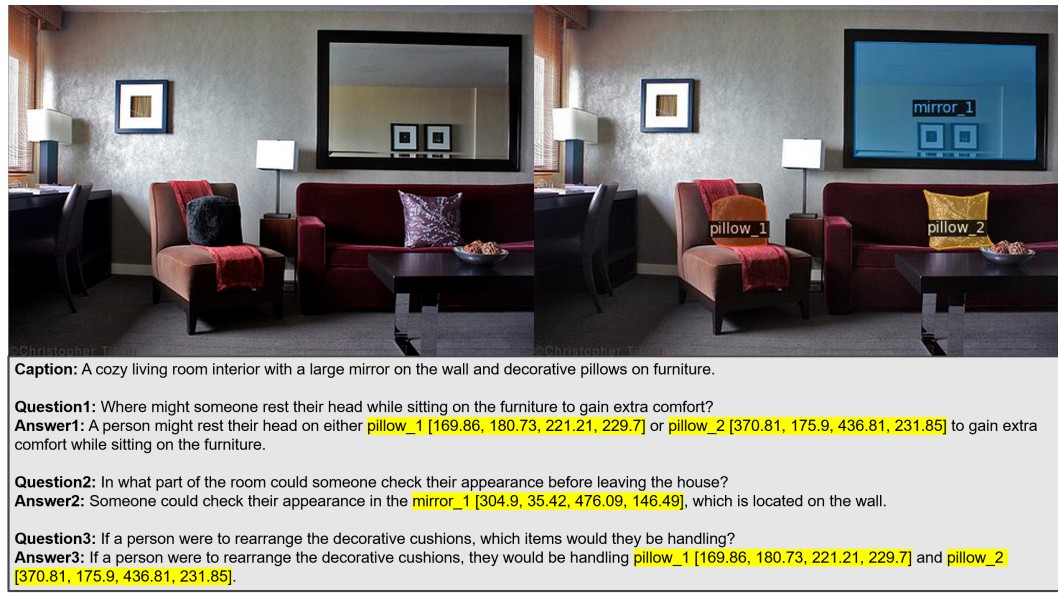

Figure 10: An example from the object-only MMR test dataset generated through the prompt in Fig. 9. This example includes information of objects. The left and right pictures show the original image and object-level segmentation masks, respectively.

```
data:
    {
        `file_name`: str, a file name of the image,
        `height`: int, height of the image,
        `width`: int, width of the image,
        `image_id`: int, id of the image,
        `not_exhaustive_category_ids`: List[int], list of category ids
        that don't have all of their instances marked exhaustively,
        `neg_category_ids`: List[int], list of category ids that were verified as not present in the image,
        `coco_url`: str, image URL,
        `questions`: List[str], the complex and implicit questions about the objects and parts in an image,
        `annotations`:
            {
              `bbox`: List[float], bounding box of the object or part,
              `segmentation`:
                  {
                    `size`: List[int], the size of the image,
                    `counts`: RLE format, segmentation binary mask information,
                  }
              `image_id`: int, id of the image,
              `category_name`: str, category_name of the object or part,
              `category_id`: int, category_id,
              `sorted_category_id`: int, sorted id in ascending order,
            }
        `answers`: List[dicts], the annotations corresponding to the questions,
        `text_answers`: List[str], the text answers to the questions,
        `raw_answers`: List[str], the raw answers from GPT API to the questions,
    }
```

Figure 11: MMR dataset format

Table 7: Comparison of computational complexity on LISA, GSVA, and GLaMM, and M$^2$SA.

| Methods | GPU Memory Usage (GB) | TFLOPs |
|---|---|---|
| LISA-7B (Lai et al., 2023) | 30.58 | 32.59 |
| GSVA-7B (Xia et al., 2024) | 30.39 | 203.77 |
| GLaMM (Rasheed et al., 2024) | 17.14 | 349.28 |
| **M$^2$SA-7B** | 30.60 | 32.62 |
| LISA-Llama2-13B (Lai et al., 2023) | 55.20 | 56.64 |
| **M$^2$SA-Llama2-13B** | 55.23 | 56.67 |

## A.8 TRAINING TIME

The training takes approximately 40 hours for the M$^2$SA-7B and about 52 hours for the M$^2$SA-Llama2-13B, respectively.

## A.9 COMPUTATIONAL COMPLEXITY

We aim to compare the computational complexity of the proposed M$^2$SA with LISA, GSVA, and GLaMM. For this comparison, we measure GPU memory usage and TFLOPs. As shown in Tab. 7, while the addition of Early Local Feature Fusion and multiple [SEG] tokens leads to a slight increase in GPU memory usage and TFLOPs, M$^2$SA demonstrates a significant improvement in handling multiple targets and fine-grained part-level segmentation compared to LISA. However, despite these performance improvements, there is still room for enhancement from the perspective of computational efficiency. Since M$^2$SA is built upon both MLLM and SAM, it requires substantial memory resources. Future research could focus on optimizing the efficiency of the mask decoder, which predicts the final mask by integrating vision and language information.

Table 8: Multi-object referring segmentation results on GTAV and Cityscapes validation sets. We adopt mIoU metric for comparison. We evaluate the zero-shot performance of LISA, GSVA, GLaMM, and M$^2$SA. The best results are highlighted in **bold**.

| Methods | GTAV-val | Cityscapes-val |
|---|---|---|
| LISA-7B (Lai et al., 2023) | 3.7 | 6.1 |
| GSVA-7B (Xia et al., 2024) | 15.7 | 14.6 |
| GLaMM (Rasheed et al., 2024) | 12.6 | 12.6 |
| **M$^2$SA-7B** | **35.1** | **41.3** |
| LISA-Llama2-13B (Lai et al., 2023) | 2.4 | 3.4 |
| **M$^2$SA-Llama2-13B** | **38.2** | **44.0** |

## A.10 GENERALIZATION ON UNSEEN DATA

To assess M$^2$SA's generalization to unseen data, we conduct additional experiments. Although OV-PARTS (Wei et al., 2024) was recently proposed for open-vocabulary part-level segmentation using Pascal-Part (Chen et al., 2014) and ADE20K (Zhou et al., 2019), both datasets were used during training. Therefore, we evaluate the model's generalization performance using semantic segmentation datasets from driving scenes, specifically Cityscapes (Cordts et al., 2016) and GTAV (Richter et al., 2016), which were not used during training and pose a more challenging test environment. Since these datasets lack part-level mask annotations, we focus on evaluating multi-target object cases. Furthermore, we curate custom text prompts using predefined category names as they do not provide corresponding text queries. For each query, we randomly select 4 to 6 object categories from an image and create prompts such as *"Can you segment the class 1, class 2, . . . , and class n?"*. The model generates masks for the specified objects, and we compute the mIoU score to compare its performance with LISA. As shown in Tab. 8, M$^2$SA performs robustly even on datasets from entirely different domains. Notably, while the existing methods struggle with multi-target cases, M$^2$SA handles them effectively. This demonstrates that the use of multiple [SEG] tokens, combined

Table 9: Comparison between LISA-7B (Lai et al., 2023) trained on MMR dataset and LISA-7B trained on ReasonSeg (Lai et al., 2023). We measure the performance on ReasonSeg validation set

| Methods | gIoU | cIoU |
|---|---|---|
| LISA-7B w/ ReasonSeg | 44.4 | 46.0 |
| LISA-7B w/ MMR | 49.9 | 55.6 |

with early local feature fusion, enables $M^2SA$ to generalize well to unseen domains by improving its ability to manage multi-target cases and fine-grained segmentation tasks.

## A.11 MMR AND REASONSEG

To validate the comprehensiveness and effectiveness of the MMR dataset, we conduct a comparative evaluation with ReasonSeg using the LISA-7B model. Specifically, we train the model in two configurations: one using ReasonSeg and the other using MMR instead of ReasonSeg. As shown in Tab. 9, the model trained on MMR shows superior performance on the ReasonSeg validation set than the model trained on ReasonSeg. This improvement highlights the comprehensiveness of the MMR dataset. By incorporating multi-target and part-level annotations alongside object-level data, MMR provides a more robust knowledge for addressing complex reasoning segmentation tasks.

Table 10: Performance of M2SA on frequently appearing and infrequently appearing object categories. From the total of 75 categories, question-answer pairs containing the top 10 most frequent (upper) and bottom 10 least frequent (lower) categories are extracted to construct the upper and lower subsets, respectively.

| Methods | MMR test | | | | | |
|---|---|---|---|---|---|---|
| | Obj-only (total) | | Obj-only (upper) | | Obj-only (lower) | |
| | gIoU | cIoU | gIoU | cIoU | gIoU | cIoU |
| **$M^2SA$-7B** | 41.0 | 55.6 | 41.0 | 54.8 | 39.4 | 39.7 |

Table 11: Performance of M2SA on frequently appearing and infrequently appearing part categories. From the total of 445 categories, question-answer pairs containing the top 10 most frequent (upper) and bottom 10 least frequent (lower) categories are extracted to construct the upper and lower subsets, respectively.

| Methods | MMR test | | | | | |
|---|---|---|---|---|---|---|
| | Part-only (total) | | Part-only (upper) | | Part-only (lower) | |
| | gIoU | cIoU | gIoU | cIoU | gIoU | cIoU |
| **$M^2SA$-7B** | 13.5 | 27.0 | 12.8 | 24.8 | 13.3 | 28.1 |

## A.12 ANALYSIS OF THE LONG-TAIL PHENOMENON IN M2SA

To investigate whether $M^2SA$ trained on the MMR dataset exhibits a long-tail phenomenon, we evaluate its performance on frequently and infrequently occurring object and part categories. To this end, we construct subsets of the MMR test set by isolating question-answer pairs based on category frequency. Specifically, we extract the top 10 most frequent (upper) and bottom 10 least frequent (lower) categories for both object-only and part-only test sets. This results in four subsets: object-only (upper: 10/75), object-only (lower: 10/75), part-only (upper: 10/445), and part-only (lower: 10/445). The MMR dataset includes a total of 75 object categories and 445 part categories, respectively. The performance comparison is shown in Tab. 10 and Tab. 11.

For the object-only dataset, $M^2SA$'s performance on frequently occurring (upper) object categories closely aligns with its overall performance across all object categories (gIoU: 41.0, cIoU: 54.8 vs. gIoU: 41.0, cIoU: 55.6). However, for infrequent object categories (lower), the performance declines, with cIoU dropping from 55.6 to 39.7 and gIoU from 41.0 to 39.4. In contrast, for the

part-only dataset, M$^2$SA demonstrates consistent performance across both frequent and infrequent categories. The gIoU scores are 12.8 (upper), 13.3 (lower), and 13.5 (overall), while the cIoU scores are 24.8 (upper), 28.1 (lower), and 27.0 (overall). These findings suggest that M$^2$SA is less sensitive to the long-tail distribution in part categories than in object categories.

This analysis highlights the strengths and limitations of M$^2$SA when addressing long-tail distributions. While M$^2$SA demonstrates robust performance across frequent and infrequent part categories, its reduced performance on infrequent object categories indicates potential areas for improvement. Future work could explore strategies to mitigate the impact of long-tail distributions in object categories while preserving its strengths in part-level reasoning tasks.

### A.13 QUALITATIVE RESULTS

Qualitative results of M$^2$SA on the MMR benchmark are visualized in Fig. 12, Fig. 13, and Fig. 14.

**Image**           **Output**

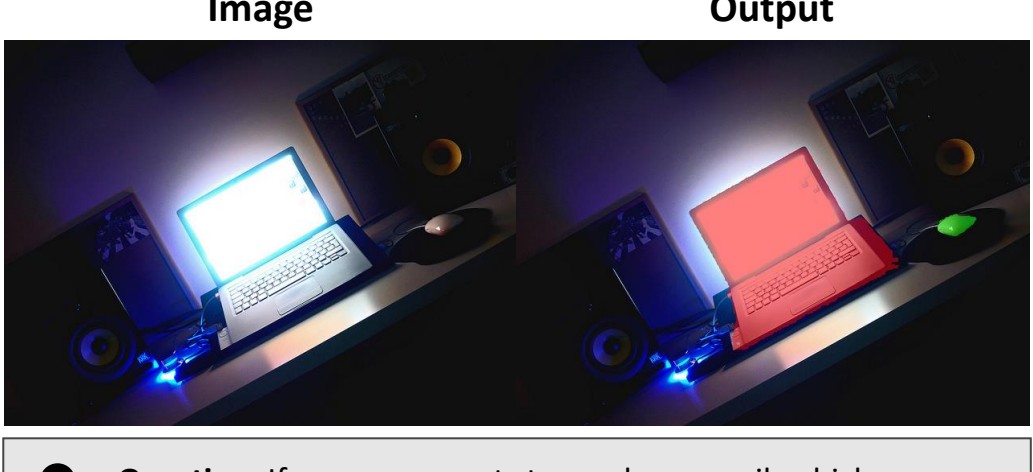

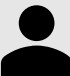 **Question:** If someone wants to send an e-mail, which equipments on the desk would they be utilizing?

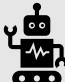 **Answer:** They would be utilizing **the laptop_computer** for typing and viewing the screen, and **the mouse** for navigating and interacting with the computer.

Figure 12: Qualitative result of M$^2$SA on MMR test set.

**Image**                    **Output**

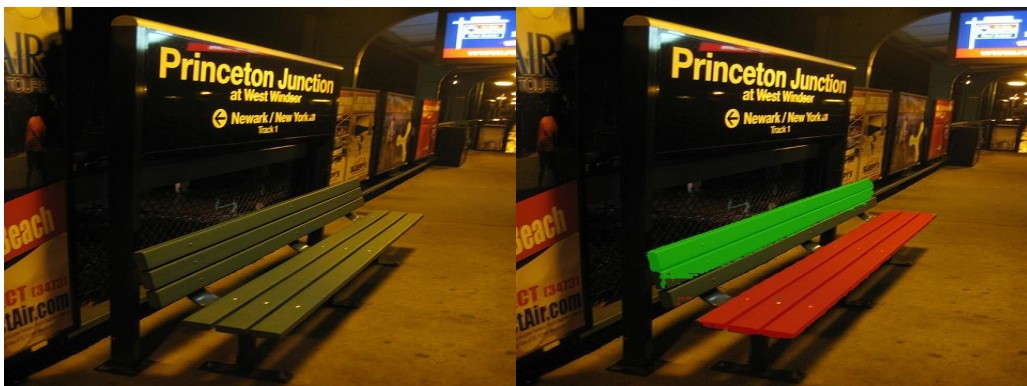

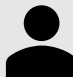 **Question:** Where could someone sit while waiting for transportation, and which part provides support for their back?

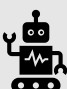 **Answer:** **The bench's seat** provides a place to sit, and **the bench's back** offers support for the back

Figure 13: Qualitative result of M$^2$SA on MMR test set.

**Image**                    **Output**

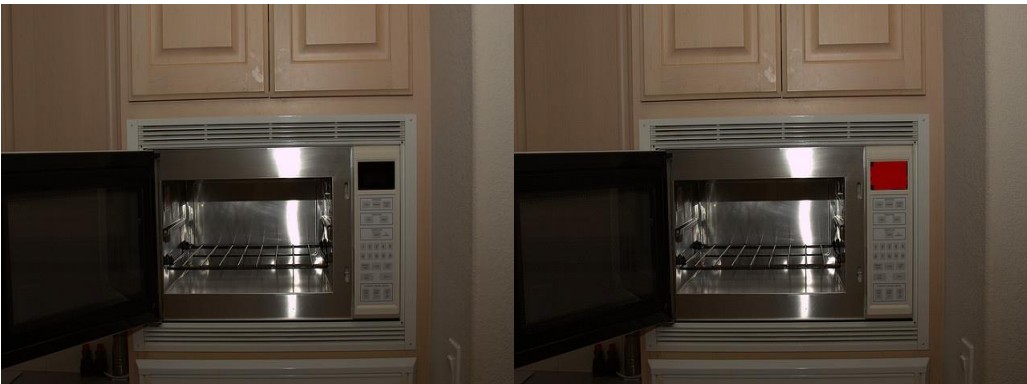

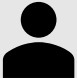 **Question:** If I need to check how much time is left before my meal is ready, which part of this appliance should I look at?

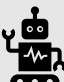 **Answer:** You should look at the **microwave_oven's time_display** to check the remaining time.

Figure 14: Qualitative result of M$^2$SA on MMR test set.

## A.14 ADDITIONAL EXAMPLES OF MMR

To facilitate a quick and intuitive understanding of the MMR dataset's characteristics, we present additional examples in Figure 15.

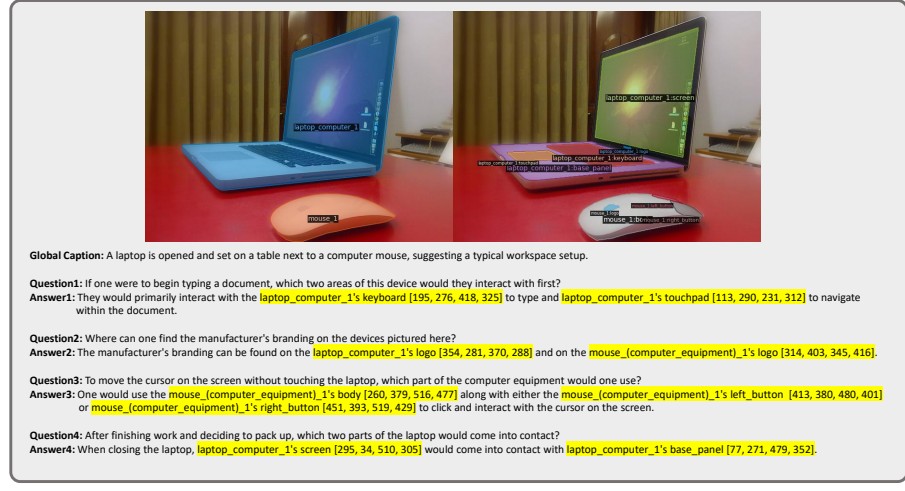

**Global Caption:** A laptop is opened and set on a table next to a computer mouse, suggesting a typical workspace setup.

**Question1:** If one were to begin typing a document, which two areas of this device would they interact with first?
**Answer1:** They would primarily interact with the laptop_computer_1's keyboard [195, 276, 418, 325] to type and laptop_computer_1's touchpad [113, 290, 231, 312] to navigate within the document.

**Question2:** Where can one find the manufacturer's branding on the devices pictured here?
**Answer2:** The manufacturer's branding can be found on the laptop_computer_1's logo [354, 281, 370, 288] and on the mouse_(computer_equipment)_1's logo [314, 403, 345, 416].

**Question3:** To move the cursor on the screen without touching the laptop, which part of the computer equipment would one use?
**Answer3:** One would use the mouse_(computer_equipment)_1's body [260, 379, 516, 477] along with either the mouse_(computer_equipment)_1's left_button [413, 380, 480, 401] or mouse_(computer_equipment)_1's right_button [451, 393, 519, 429] to click and interact with the cursor on the screen.

**Question4:** After finishing work and deciding to pack up, which two parts of the laptop would come into contact?
**Answer4:** When closing the laptop, laptop_computer_1's screen [295, 34, 510, 305] would come into contact with laptop_computer_1's base_panel [77, 271, 479, 352].

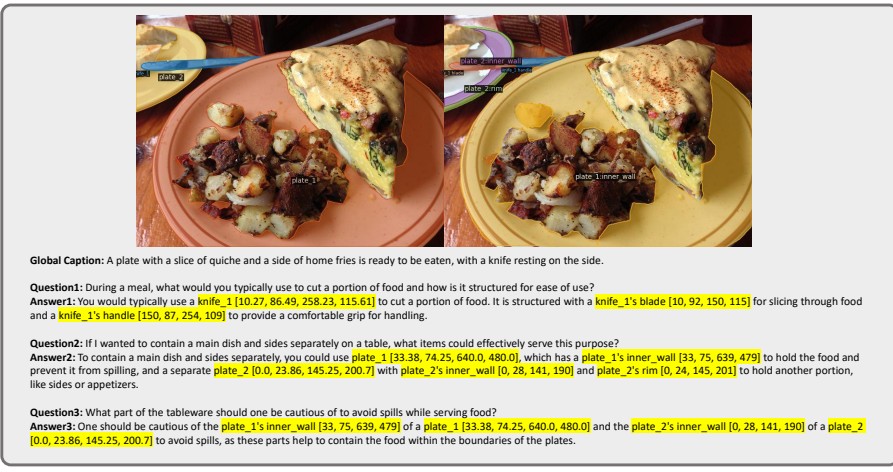

**Global Caption:** A plate with a slice of quiche and a side of home fries is ready to be eaten, with a knife resting on the side.

**Question1:** During a meal, what would you typically use to cut a portion of food and how is it structured for ease of use?
**Answer1:** You would typically use a knife_1 [10.27, 86.49, 258.23, 115.61] to cut a portion of food. It is structured with a knife_1's blade [10, 92, 150, 115] for slicing through food and a knife_1's handle [150, 87, 254, 109] to provide a comfortable grip for handling.

**Question2:** If I wanted to contain a main dish and sides separately on a table, what items could effectively serve this purpose?
**Answer2:** To contain a main dish and sides separately, you could use plate_1 [33.38, 74.25, 640.0, 480.0], which has a plate_1's inner_wall [33, 75, 639, 479] to hold the food and prevent it from spilling, and a separate plate_2 [0.0, 23.86, 145.25, 200.7] with plate_2's inner_wall [0, 28, 141, 190] and plate_2's rim [0, 24, 145, 201] to hold another portion, like sides or appetizers.

**Question3:** What part of the tableware should one be cautious of to avoid spills while serving food?
**Answer3:** One should be cautious of the plate_1's inner_wall [33, 75, 639, 479] of a plate_1 [33.38, 74.25, 640.0, 480.0] and the plate_2's inner_wall [0, 28, 141, 190] of a plate_2 [0.0, 23.86, 145.25, 200.7] to avoid spills, as these parts help to contain the food within the boundaries of the plates.

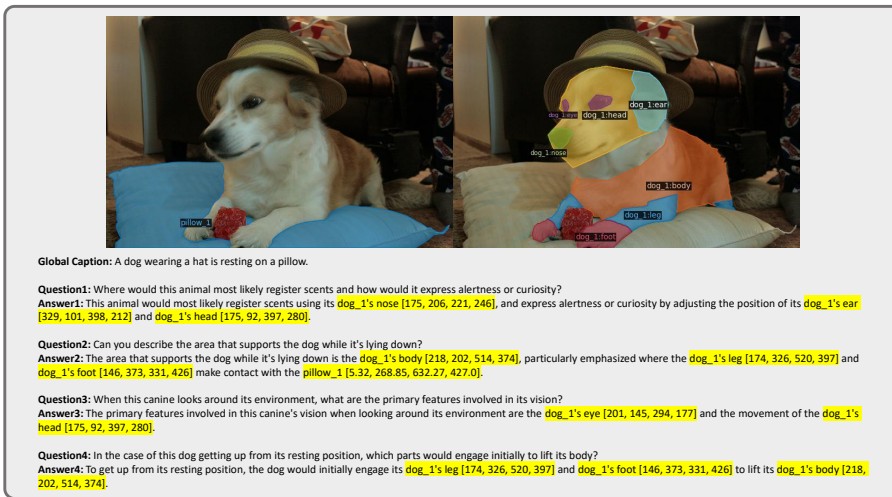

**Global Caption:** A dog wearing a hat is resting on a pillow.

**Question1:** Where would this animal most likely register scents and how would it express alertness or curiosity?
**Answer1:** This animal would most likely register scents using its dog_1's nose [175, 206, 221, 246], and express alertness or curiosity by adjusting the position of its dog_1's ear [329, 101, 398, 212] and dog_1's head [175, 92, 397, 280].

**Question2:** Can you describe the area that supports the dog while it's lying down?
**Answer2:** The area that supports the dog while it's lying down is the dog_1's body [218, 202, 514, 374], particularly emphasized where the dog_1's leg [174, 326, 520, 397] and dog_1's foot [146, 373, 331, 426] make contact with the pillow_1 [5.32, 268.85, 632.27, 427.0].

**Question3:** When this canine looks around its environment, what are the primary features involved in its vision?
**Answer3:** The primary features involved in this canine's vision when looking around its environment are the dog_1's eye [201, 145, 294, 177] and the movement of the dog_1's head [175, 92, 397, 280].

**Question4:** In the case of this dog getting up from its resting position, which parts would engage initially to lift its body?
**Answer4:** To get up from its resting position, the dog would initially engage its dog_1's leg [174, 326, 520, 397] and dog_1's foot [146, 373, 331, 426] to lift its dog_1's body [218, 202, 514, 374].

Figure 15: Additional Examples of MMR dataset.

