# OpenReview forum: "MMR: A Large-scale Benchmark Dataset for Multi-target and Multi-granularity Reasoning Segmentation"
_ICLR.cc/2025/Conference — ICLR 2025 Poster_

### Official Review · Reviewer_D24r · 2024-10-22

**Soundness:** 3
**Presentation:** 4
**Contribution:** 3
**Rating:** 6
**Confidence:** 5

**Summary:**

This paper introduces a novel dataset, MMR, the first part-level dataset for the reasoning segmentation tasks. In addition, a new network framework is proposed to leverage low-level fine-grained information and to address the limitation of the existing LISA model, which can only segment a single object. The authors conduct experiments to evaluate the performance of existing methods on the proposed MMR dataset and demonstrate the advantages of the proposed network framework.

**Strengths:**

1. This paper is well written and easy to follow.
2. The proposed MMR dataset is highly valuable to the research community, as part-level reasoning segmentation is crucial in real-world applications, such as robotic control. However, there is currently a lack of available datasets for research in this area.
3. A detailed analysis is provided to thoroughly present the characteristics of the MMR dataset.

**Weaknesses:**

1. The contributions of the proposed M2SA network framework are incremental. The early local feature fusion appears to be only a minor structural modification. Additionally, the strategy of employing multiple [SEG] tokens has already been introduced in earlier methods, such as [a]. The authors should clarify the differences between their approach and [a].
2. This paper could benefit from more thorough experiments based on the characteristics of the dataset. For instance, does the M2SA trained on the MMR dataset show a noticeable long-tail phenomenon, i.e., better performance on the more frequently occurring object and part categories as presented in Fig.3? Additionally, what is the model’s open-vocabulary performance on categories that do not appear in the training set?
3. More examples of the image-question-answer triplet in the MMR dataset could be presented in the paper to enable readers to understand the characteristics of the dataset more quickly and intuitively.

[a] GSVA: Generalized Segmentation via Multimodal Large Language Models, CVPR2024

**Questions:**

1. Why did you remove the generated questions that contain explicit target coordinates or strong hints? I think training with such data would enhance the model’s ability to handle target-specific inputs. For example, if an image contains two different animals, a fish and a cat, users could indicate the coordinates of the animal they are interested in and ask, “Which part of this animal [coordinates] uses its sense of smell?” The model could then segment either the nose or the fish’s gills depending on the coordinates provided. This could be quite interesting.

---

> ### Author Response · Authors · 2024-11-22
> **Author's response (1/4)**
>
> We would like to express our gratitude for the time you dedicated to providing insightful feedback on our work. We are pleased to hear that you found our dataset useful and intriguing in the context of the reasoning segmentation task. Following the reviewer's feedback, we will update our manuscript, clarifying the difference from previous works. Please let us know if any aspects remain unclear or if you have additional comments. We would be very grateful for any further feedback and will address any additional inputs promptly and thoroughly.
>
>
> > The contributions of the proposed M2SA network framework are incremental. The early local feature fusion appears to be only a minor structural modification. Additionally, the strategy of employing multiple [SEG] tokens has already been introduced in earlier methods, such as [a]. The authors should clarify the differences between their approach and [a].
>
> We appreciate the reviewer’s insightful comments. While GSVA [a] and the proposed framework (M2SA) share the use of multiple [SEG] tokens, there is a difference in the information embedded in the multiple [SEG] tokens during the learning process. While GSVA employs multiple [SEG] tokens for multi-target object-level segmentation, the proposed framework allows the [SEG] tokens to embed hierarchical information that differentiates objects and their parts during training. This is achieved through Early Local Feature Fusion which combines fine-grained local features with global semantic features.
>
> Consequently, the proposed framework embeds richer information about spatial interactions into the [SEG] tokens. For example, the proposed framework can generate distinct [SEG] tokens for "dog's ear" and "dog's nose," effectively capturing part-level hierarchies and relationships that GSVA cannot. These enhancements enable the proposed framework to excel in tasks requiring multi-granularity reasoning segmentation, as demonstrated in the comparison on the MMR dataset.
>
> Table. The performance comparison between GSVA and the proposed framework (M2SA) on the MMR dataset. GSVA-7B is the pre-trained model provided by the authors. GSVA-7B_tr is the trained model from scratch on a mixed training set that includes the MMR dataset.
> | Methods | val (gIoU) | val (cIoU) | Object-only (gIoU) | Object-only (cIoU) | Part-only (gIoU) | Part-only (gIoU) |Object & Part (gIoU) | Object & Part  (gIoU)  |
> |:------|:-------------:|:-------------:|:-----------:|:-------------:|:---------------:|:-------------:|:----------------:|:---------------------:|
> |GSVA-7B|14.6|25.1|26.4|34.3|6.0|11.6|15.5|24.8|
> |GSVA-7B_tr |19.8|38.9|30.2|41.1|8.0|18.6|21.2|34.5|
> |**M2SA-7B**|27.9|48.5|41.0|55.6|13.5|27.0|31.0|46.8|
>
>
>
> [a] Xia, Z., Han, D., Han, Y., Pan, X., Song, S., & Huang, G. (2024). Gsva: Generalized segmentation via multimodal large language models. In Proceedings of the IEEE/CVF Conference on Computer Vision and Pattern Recognition (pp. 3858-3869).

---

> ### Author Response · Authors · 2024-11-22
> **Author's response (2/4)**
>
> > Does the M2SA trained on the MMR dataset show a noticeable long-tail phenomenon, i.e., better performance on the more frequently occurring object and part categories as presented in Fig.3?
>
>
> Thank you for your insightful comment on examining the effects based on the frequency of category appearances. To verify whether the long-tail phenomenon is observed, we constructed subsets of the MMR test set by isolating question-answer pairs for frequently and infrequently occurring categories. Specifically, we extracted the top 10 most frequent (upper) and bottom 10 least frequent (lower) categories for both object-only and part-only datasets This resulted in four subsets: object-only (upper: 10/75), object-only (lower: 10/75), part-only (upper: 10/445), and part-only (lower: 10/445).  (MMR includes a total of 75 object categories and 445 part categories, respectively.)
>
> For the object-only dataset, we observe that the performance on frequently occurring (upper) categories is similar to the overall performance, while the cIoU score for infrequent (lower) categories is lower. In contrast, for the part-only dataset, no significant differences are observed between the upper and lower subsets compared to the performance on all the categories.
>
>
> Table. Performance of M2SA on frequently appearing and infrequently appearing object categories. From the total of 75 categories, question-answer pairs containing the top 10 most frequent (upper) and bottom 10 least frequent (lower) categories were extracted to construct the upper and lower subsets, respectively.
> | Methods | Object-only (total), gIoU | Object-only (total), cIoU | Object-only (upper), gIoU | Object-only (upper), cIoU | Object-only (lower), gIoU | Object-only (lower), cIoU |
> |:------|:-------------:|:-------------:|:-----------:|:-------------:|:-----------:|:-------------:|
> |**M2SA-7B**|41.0|55.6|41.0|54.8|39.4|39.7|
>
>  Table. Performance of M2SA on frequently appearing and infrequently appearing part categories. From the total of 445 categories, question-answer pairs containing the top 10 most frequent (upper) and bottom 10 least frequent (lower) categories were extracted to construct the upper and lower subsets, respectively.
> | Methods | Part-only (total), gIoU | Part-only (total), cIoU | Part-only (upper), gIoU | Part -only (upper), cIoU | Part-only (lower), gIoU | Part-only (lower), cIoU |
> |:------|:-------------:|:-------------:|:-----------:|:-------------:|:-----------:|:-------------:|
> |**M2SA-7B**|13.5|27.0|12.8|24.8|13.3|28.1|

---

> ### Author Response · Authors · 2024-11-22
> **Author's response (3/4)**
>
> >Additionally, what is the model’s open-vocabulary performance on categories that do not appear in the training set?
>
>
> Thank you for your insightful suggestion. While OV-PARTS [1] was recently proposed for open-vocabulary part-level segmentation, its dataset is based on Pascal-Part and ADE20K, which are also utilized during model training, limiting its appropriateness for evaluation. Instead, we aim to measure open-vocabulary performance using segmentation datasets related to driving scenes, such as Cityscapes [2] and GTAV [3]. These datasets are not included in the training data and have a significant domain gap from the training datasets, making them a challenging scenario.
>
> However, since these datasets do not provide part-level mask annotations, we focus on evaluating multi-target object cases. Additionally, these datasets lack text queries corresponding to the images and masks. To address this, we curate text queries specifically for multi-target object cases using the category names in the datasets. For multi-target queries, we randomly select 4 to 6 object categories from each image and construct queries like: "Can you segment the {class 1, class 2, …, and class n}?" The model then generates masks for the listed objects in response to these queries. We calculate the mIoU score and compare the performance of LISA, GSVA [4], GLaMM [5], and our proposed M$^{2}$SA.
>
> Table. Multi-object referring segmentation results on GTAV and Cityscapes validation sets. We adopt the mIoU metric for comparison. We evaluate the zero-shot performance of the LISA, GSVA, GLaMM, and M$^{2}$SA.
> |Methods|GTAV-val|Cityscapes-val|
> |:----|:---:|:----:|
> |LISA-7B|3.7|6.1|
> |GSVA-7B|15.7|14.6|
> |GLaMM |12.6|12.6|
> |**M2SA-7B**|35.1|41.3|
>  We observe that the proposed model (M2SA) achieves strong performance even on datasets from entirely different domains. Notably, while existing methods struggle with multi-target cases, the proposed model (M2SA) handles them effectively. This result indicates that incorporating multiple [SEG] tokens for handling multi-target cases and enhancing fine-grained understanding through early local feature fusion enables the model to generalize well to unseen domain datasets.
>
> [1] Wei, M., Yue, X., Zhang, W., Kong, S., Liu, X., & Pang, J. (2024). Ov-parts: Towards open-vocabulary part segmentation. Advances in Neural Information Processing Systems, 36.
> [2] Cordts, M., Omran, M., Ramos, S., Rehfeld, T., Enzweiler, M., Benenson, R., ... & Schiele, B. (2016). The cityscapes dataset for semantic urban scene understanding. In Proceedings of the IEEE conference on computer vision and pattern recognition (pp. 3213-3223).
> [3] Richter, S. R., Vineet, V., Roth, S., & Koltun, V. (2016). Playing for data: Ground truth from computer games. In Computer Vision–ECCV 2016: 14th European Conference, Amsterdam, The Netherlands, October 11-14, 2016, Proceedings, Part II 14 (pp. 102-118). Springer International Publishing.
> [4] Xia, Z., Han, D., Han, Y., Pan, X., Song, S., & Huang, G. (2024). Gsva: Generalized segmentation via multimodal large language models. In Proceedings of the IEEE/CVF Conference on Computer Vision and Pattern Recognition (pp. 3858-3869).
> [5] Rasheed, H., Maaz, M., Shaji, S., Shaker, A., Khan, S., Cholakkal, H., ... & Khan, F. S. (2024). Glamm: Pixel grounding large multimodal model. In Proceedings of the IEEE/CVF Conference on Computer Vision and Pattern Recognition (pp. 13009-13018).

---

> ### Author Response · Authors · 2024-11-22
> **Author's response (4/4)**
>
> > More examples of the image-question-answer triplet in the MMR dataset could be presented in the paper to enable readers to understand the characteristics of the dataset more quickly and intuitively.
>
> Thank you for your valuable feedback. Following the reviewer’s suggestion, we included more examples in Appendix A.11 of the updated manuscript.
>
> > Why did you remove the generated questions that contain explicit target coordinates or strong hints? I think training with such data would enhance the model’s ability to handle target-specific inputs. For example, if an image contains two different animals, a fish and a cat, users could indicate the coordinates of the animal they are interested in and ask, “Which part of this animal [coordinates] uses its sense of smell?” The model could then segment either the nose or the fish’s gills depending on the coordinates provided. This could be quite interesting.
>
>
> Thank you for your valuable comment. The task you mentioned is closely related to the "grounding" task. However, our MMR dataset is designed with a different focus: creating a reasoning segmentation dataset that emphasizes addressing implicit queries rather than explicit ones. For example, consider a query like, "Which part of this animal [coordinates] uses its sense of smell?" If the target coordinates are explicitly provided, the task shifts away from the core objective of a reasoning segmentation task. Instead of inferring which part of the animal’s body is used for smelling, the model simply identifies the corresponding part near the given coordinates in the image. This aligns more closely with the "grounding" task, which focuses on explicit localization rather than reasoning. In contrast, our MMR dataset is specifically designed to require deeper reasoning based on implicit queries to acquire the correct segmentation mask. To align with this objective, we have intentionally filtered out such explicit cases, ensuring the dataset remains focused on fostering deeper reasoning capabilities.

---

> > ### Comment · Reviewer_D24r · 2024-11-23
> > **Thanks for the response**
> >
> > Thank the authors for the detailed response. While I still think the improvements to the network structure are incremental, the authors have provided additional comparisons with GSVA in the rebuttal, and the results demonstrate significant advantages. Furthermore, the proposed benchmark holds valuable potential for the research community. As such, I have decided to maintain my original score of acceptance.

---

### Official Review · Reviewer_JQ5y · 2024-10-22

**Soundness:** 3
**Presentation:** 3
**Contribution:** 3
**Rating:** 6
**Confidence:** 3

**Summary:**

This paper introduces a dataset named MMR, designed for multi-target and multi-granularity reasoning segmentation tasks. The goal is to address challenges in reasoning across multiple targets and different levels of granularity. The dataset comprises complex and implicit question pairs, covering both object-level and part-level reasoning. Additionally, the paper proposes a baseline model, M2SA, to achieve multi-target, object-level, and part-level reasoning segmentation.

**Strengths:**

1. **Clear Writing**: The paper is well-organized and easy to understand.

2. **Significant Contribution of the Dataset**: The MMR dataset contains 196K samples. Although it was generated using large models, a rigorous filtering process was employed to ensure data quality.

**Weaknesses:**

1. **Lack of Targeted Design in the Baseline Model**: The baseline model (Early Local Feature Fusion and Multi [SEG] Tokens) does not incorporate specific structures to effectively address the multi-target and part-level reasoning required by the MMR dataset. As a result, it lacks novelty, leading to underwhelming performance in Table 3.

2. **Limited Performance in Table 3**: The comparison methods in Table 3 are not sufficiently recent. The authors did not include comparisons with more relevant multi-target approaches, such as GSVA [1] or GLaMM [2]. This limits the impact of the proposed approach, as the results are not very competitive.

3. **Insufficient Comparisons in Table 2**: The methods compared in Table 2 are too limited. I strongly recommend including more methods that could be adapted for the MMR task to facilitate meaningful comparisons for future research.

**Questions:**

1. Will the proposed dataset be made publicly available?
2. The paper mentions the use of 4 A6000 GPUs. How long does it take to train the proposed model on the MMR dataset?

---

> ### Author Response · Authors · 2024-11-22
> **Author's response (1/2)**
>
> Thank you for taking the time to evaluate our paper and for providing valuable suggestions. We are pleased that you found the significant contribution of our proposed dataset. In response to your feedback, we will update the manuscript accordingly to address your suggestions and improve the overall quality of our work. We appreciate your insights and believe these revisions will strengthen our submission. If any aspects of our response that are unclear or if you have any further questions, please let us know.
>
>
> Point-by-point response:
> > Lack of Targeted Design in the Baseline Model: The baseline model (Early Local Feature Fusion and Multi [SEG] Tokens) does not incorporate specific structures to effectively address the multi-target and part-level reasoning required by the MMR dataset. As a result, it lacks novelty, leading to underwhelming performance in Table 3.
>
>
> Thank you for your thoughtful feedback. The proposed Multi [SEG] Tokens and Early Local Feature Fusion mechanisms are specifically introduced to address the multi-target and part-level reasoning requirements of the MMR dataset. Multi [SEG] Tokens enable the model to dynamically adapt to varying numbers of objects in a query, directly tackling the multi-target scenarios. Early Local Feature Fusion enhances the model’s ability to capture fine-grained part-level details and hierarchical relationships by considering spatial information.
>
> It is worth noting that our method is designed for reasoning segmentation, whereas Table 3 evaluates performance on referring expression segmentation tasks. Referring expression segmentation typically involves explicit, single-target text queries (e.g., "right horse"), which are simpler and less demanding than the implicit, multi-target queries emphasized in reasoning segmentation. This distinction explains the relatively modest performance gains observed in Table 3.
>
> To evaluate the capability of existing methods in more challenging scenarios, we perform comparisons on multi-target referring expression segmentation. For this evaluation, we curate multi-target text queries using annotation information from the RefCOCO-series datasets. Specifically, we randomly selected 4 to 6 object categories from each image and constructed text queries such as: "Can you segment the {class 1, class 2, …, and class n}?"
>
> Table. Multi-referring expression segmentation results. We adopt the cIoU metric for a comparison
> | Methods | Multi-RefCOCO (val) | Multi-RefCOCO (testA) | Multi-RefCOCO (testB) | Multi-RefCOCO+ (val) | Multi-RefCOCO+ (testA) | Multi-RefCOCO+ (testB) | Multi-RefCOCOg (val(U)) | Multi-RefCOCOg (test(U)) |
> |:------|:-------------:|:-------------:|:-----------:|:-------------:|:---------------:|:-------------:|:----------------:|:---------------------:|
> |LISA-7B|34.0|32.7|34.5|28.2|28.6|28.5|45.2|48.7|
> |GSVA-7B|50.7|53.3|47.8|44.8|47.4|40.6|47.7|48.6|
> |GLaMM|30.8|32.0|30.0|28.8|29.6|27.2|32.5|35.0|
> |**M2SA-7B**|71.6|73.6|67.6|61.0|64.3|55.8|61.8|63.3|
>
> These results show that M2SA achieves significant improvement across all multi-target referring segmentation datasets. This underscores the advantages of incorporating multiple [SEG] tokens and early local feature fusion for handling multi-target cases. We included these results in Table 4 of the updated manuscript. Thank you for your valuable insights on strengthening our approach.
>
> > Limited Performance in Table 3: The comparison methods in Table 3 are not sufficiently recent. The authors did not include comparisons with more relevant multi-target approaches, such as GSVA [1] or GLaMM [2]. This limits the impact of the proposed approach, as the results are not very competitive.
> sdsdsds
>
>
> Thank you for highlighting the importance of including comparisons with more recent multi-target approaches like GSVA and GLaMM. We fully agree that incorporating these methods is essential for a comprehensive evaluation of our proposed approach. At the time of conducting this work, GSVA's code was not publicly available, which prevented us from including its results initially. However, based on the reviewer’s suggestion, we have now evaluated GSVA and GLaMM on both the single-target referring expression segmentation task (Table 3 in the updated manuscript), the multi-target referring expression segmentation task (Table 4 in the updated manuscript), and the multi-granularity referring expression segmentation (Table 5 in the updated manuscript).
>
> As noted in concern 1, while the proposed method shows marginal performance improvements in single-target referring expression segmentation compared to existing methods, it achieves superior performance in the more challenging scenario of multi-target referring expression segmentation and multi-granularity referring expression segmentation. These additional results further highlight the competitiveness of M2SA compared to recent multi-target approaches.

---

> ### Author Response · Authors · 2024-11-22
> **Author's response (2/2)**
>
> > Insufficient Comparisons in Table 2: The methods compared in Table 2 are too limited. I strongly recommend including more methods that could be adapted for the MMR task to facilitate meaningful comparisons for future research.
>
> Thank you for your thoughtful suggestion. At the time of conducting this work, the GSVA code had not been publicly released, so we were unable to include its results. We are pleased to take this opportunity to include the performance of GSVA and GLaMM on the MMR dataset. According to the authors of GSVA, the GSVA-Llama2-13B model requires 8 A100 GPUs (80GB), but due to our limited resources of 4 A6000 GPUs (40GB), we could only evaluate the 7B models. We kindly ask for your understanding regarding this limitation. The updated comparison results with additional methods on the MMR dataset are shown in the table below.
>
> Table. Results on MMR benchmark. The gIoU and cIoU metrics are reported for the comparison. Obj \& Part, Obj, and Part denote multi-granularity, object-only, and part-only evaluation settings.
> | Methods | val: Object & Part (gIoU) | val: Object & Part (cIoU) | test: Object-only (gIoU) | test: Object-only (cIoU) | test: Part-only (gIoU) |test: Part-only (gIoU) |test: Object & Part (gIoU) | test: Object & Part (cIoU)  |
> |:------|:-------------:|:-------------:|:-----------:|:-------------:|:---------------:|:-------------:|:----------------:|:---------------------:|
> |LISA-7B|13.8|18.3|23.5|25.1|6.6|7.9|14.5|17.9|
> |LISA-7B_tr|19.4|31.6|34.7|41.8|8.0|13.1|19.5|27.1|
> |GSVA-7B|14.6|25.1|26.4|34.3|6.0|11.6|15.5|24.8|
> |GSVA-7B_tr |19.8|38.9|30.2|41.1|8.0|18.6|21.2|34.5|
> |GLaMM|12.6| 19.2| 23.7| 31.9| 3.9| 6.4| 13.3| 18.7|
> |GLaMM_tr|26.9 | 47.1 | 40.3 |54.2 |12.1  |25.5  |30.3 |45.0 |
> |**M2SA-7B**|27.9|48.5|41.0|55.6|13.5|27.0|31.0|46.8|
>
> As demonstrated in the table, existing pre-trained models exhibit poor performance on the MMR benchmark, particularly in the part-only evaluation, due to their insufficient understanding of detailed part-level information. In the case of model_tr trained on the MMR dataset, we observe the performance improvement. Nevertheless, its capacity to manage intricate multi-target and part-level reasoning scenarios remains somewhat constrained. In contrast, the proposed M2SA model shows outstanding performance, handling multi-target scenarios and fine-detail tasks, underscoring its effectiveness in comprehensive reasoning segmentation. GLaMM_tr achieves competitive performance with M2SA. This is likely because, unlike LISA, GSVA, and M2SA, which are trained using LLaVA-v1.1, GLaMM leverages the pre-trained knowledge of the significantly more advanced LLaVA-v1.5.
>
> We appreciate your suggestion, as it allows us to demonstrate the effectiveness and competitiveness of our approach against more methods. We included this result in the updated version of our manuscript.
>
>
>
>
> > Will the proposed dataset be made publicly available?
>
> Yes. If our paper is accepted, we will release the proposed dataset and training code.
>
> > The paper mentions the use of 4 A6000 GPUs. How long does it take to train the proposed model on the MMR dataset?
>
> It took approximately 40 hours for the 7B model and about 52 hours for the 13B model.

---

### Official Review · Reviewer_WfTF · 2024-11-01

**Soundness:** 3
**Presentation:** 3
**Contribution:** 2
**Rating:** 6
**Confidence:** 4

**Summary:**

This paper provides a large multi-target and multi-granularity reasoning segmentation benchmark. Based on this benchmark, this paper designs a baseline model trained on it while evaluating public datasets to present the effectiveness of both the benchmarks and the baseline. Experiments demonstrate that the proposed baseline outperforms LISA and other representative approaches.

**Strengths:**

1. The distinguishing characteristic of the proposed benchmark is clear, which includes multi-granularity and more images.
2. Multi-target and multi-granularity reasoning segmentation is a valuable research topic.
3. The overall writing is fluent.

**Weaknesses:**

1. This paper provides few comparisons on the proposed benchmark. It is not clear whether the proposed baseline model outperforms other MLLMs on multi-target and multi-granularity reasoning segmentation.
2. The major contribution lies in the benchmark, while this benchmark is auto-annotated based on the existing dataset PACO-LVIS, which hurts the contribution.
3. According to Table 1, MMR offers both object-level and multi-target annotations, making it more comprehensive than ReasonSeg and MUSE. This paper could include zero-shot evaluations on these two benchmarks to further demonstrate effectiveness.

**Questions:**

Please refer to the weaknesses.

---

> ### Author Response · Authors · 2024-11-22
> **Author's response (1/2)**
>
> We appreciate the time you have invested in reviewing our work. We are very pleased that you identified the key aspects of our proposed dataset. We will update the manuscript to incorporate the suggestions you have provided. If any aspects of our response are unclear, please let us know.
>
> Point-by-point response:
> > This paper provides few comparisons on the proposed benchmark. It is not clear whether the proposed baseline model outperforms other MLLMs on multi-target and multi-granularity reasoning segmentation.
>
> Thank you for your insightful suggestion. We also agree that more comparisons with other methods on the MMR benchmark are necessary. At the time of conducting this work, the GSVA code had not been publicly released, so we were unable to include its results. We are pleased to take this opportunity to include the performance of GSVA [1] and GLaMM [2] on the MMR dataset. According to the authors of GSVA, the GSVA-Llama2-13B model requires 8 A100 GPUs (80GB), but due to our limited resources of 4 A6000 GPUs (40GB), we could only evaluate the 7B models. We kindly ask for your understanding regarding this limitation. The updated comparison results with additional baseline models on the MMR dataset are shown in the table below.
>
> Table. Results on MMR benchmark. The gIoU and cIoU metrics are reported for the comparison. Obj \& Part, Obj, and Part denote multi-granularity, object-only, and part-only evaluation settings.
> | Methods | val: Object & Part (gIoU) | val: Object & Part (cIoU) | test: Object-only (gIoU) | test: Object-only (cIoU) | test: Part-only (gIoU) |test: Part-only (gIoU) |test: Object & Part (gIoU) | test: Object & Part (cIoU)  |
> |:------|:-------------:|:-------------:|:-----------:|:-------------:|:---------------:|:-------------:|:----------------:|:---------------------:|
> |LISA-7B|13.8|18.3|23.5|25.1|6.6|7.9|14.5|17.9|
> |LISA-7B_tr|19.4|31.6|34.7|41.8|8.0|13.1|19.5|27.1|
> |GSVA-7B|14.6|25.1|26.4|34.3|6.0|11.6|15.5|24.8|
> |GSVA-7B_tr |19.8|38.9|30.2|41.1|8.0|18.6|21.2|34.5|
> |GLaMM|12.6| 19.2| 23.7| 31.9| 3.9| 6.4| 13.3| 18.7|
> |GLaMM_tr|26.9 | 47.1 | 40.3 |54.2 |12.1  |25.5  |30.3 |45.0|
> |**M2SA-7B**|27.9|48.5|41.0|55.6|13.5|27.0|31.0|46.8|
>
> As demonstrated in the table, existing pre-trained models exhibit worse performance on the MMR benchmark, particularly in the part-only evaluation, due to their insufficient understanding of detailed part-level information. In the case of model_tr trained on the MMR dataset, we observe the performance improvement. Nevertheless, its capacity to manage intricate multi-target and part-level reasoning scenarios remains somewhat constrained. In contrast, the proposed M2SA model shows outstanding performance, handling multi-target scenarios and fine-detail tasks, underscoring its effectiveness in comprehensive reasoning segmentation. GLaMM_tr achieves competitive performance with M2SA. This is likely because, unlike LISA, GSVA, and M2SA, which are trained using LLaVA-v1.1, GLaMM leverages the pre-trained knowledge of the significantly more advanced LLaVA-v1.5. We conjecture better performance compared to existing methods such as LISA and GSVA.
>
> We included this result in the updated version of our manuscript.
>
> [1] Xia, Z., Han, D., Han, Y., Pan, X., Song, S., & Huang, G. (2024). Gsva: Generalized segmentation via multimodal large language models. In Proceedings of the IEEE/CVF Conference on Computer Vision and Pattern Recognition (pp. 3858-3869).
> [2] Rasheed, H., Maaz, M., Shaji, S., Shaker, A., Khan, S., Cholakkal, H., ... & Khan, F. S. (2024). Glamm: Pixel grounding large multimodal model. In Proceedings of the IEEE/CVF Conference on Computer Vision and Pattern Recognition (pp. 13009-13018).
>
> >The major contribution lies in the benchmark, while this benchmark is auto-annotated based on the existing dataset PACO-LVIS, which hurts the contribution.
>
>
> Thank you for raising this concern. While it is true that the MMR dataset utilizes annotations from the images in the PACO-LVIS dataset, we believe it introduces a significant contribution beyond the source data by leveraging this information to create advanced reasoning segmentation text data. The proposed MMR dataset includes novel question-answer pairs addressing both object-level and part-level relationships for multi-target and multi-granularity reasoning segmentation. It is important to note that our approach goes beyond simply using the annotations in PACO-LVIS; we utilized the given image information to generate large-scale question-answer pairs (text data). These pairs were curated to include complex, implicit queries that demand reasoning capabilities beyond the annotations of PACO-LVIS. We expect the MMR dataset to contribute significantly to expanding the research scope of reasoning segmentation tasks.

---

> ### Author Response · Authors · 2024-11-22
> **Author's response (2/2)**
>
> > According to Table 1, MMR offers both object-level and multi-target annotations, making it more comprehensive than ReasonSeg and MUSE. This paper could include zero-shot evaluations on these two benchmarks to further demonstrate effectiveness.
>
> Thank you for your excellent suggestion on validating the effectiveness of the MMR dataset by comparing it with existing reasoning segmentation datasets. To evaluate the impact of ReasonSeg and MMR exclusively, we conducted evaluations using the LISA-7B model in two cases: one trained on ReasonSeg and the other trained on MMR instead of ReasonSeg. The result, presented in the table below, shows that the model trained on MMR performs better on the ReasonSeg validation set than the one trained on ReasonSeg itself. This highlights the comprehensiveness of MMR, which provides multi-target and part-level information alongside object-level data, offering a more robust knowledge for reasoning segmentation tasks.
>
> Regarding the MUSE dataset, we agree that including it in the comparison would have added valuable insights. However, we encountered several technical challenges with the PixelLM model's official code, specifically designed for training on MUSE. Specifically, as detailed in https://github.com/MaverickRen/PixelLM/issues/10, there were discrepancies in the implementation of the loss function, and an AssertionError occurred during training, as described in https://github.com/MaverickRen/PixelLM/issues/16. These issues have impacted the reproducibility of results, and despite the authors acknowledging these problems, no updates or corrective actions have been provided. To ensure reliable and consistent comparison results, we decided not to include the PixelLM model or the MUSE dataset in this analysis. We kindly ask for your understanding regarding this decision.
> Table. Performance comparison on the ReasonSeg validation set. LISA-7B trained on ReasonSeg and the LISA-7B trained on MMR instead of ReasonSeg are evaluated.
> | Methods | gIoU | cIoU |
> |:-------------------|:-------------:|:-------------:|
> |LISA-7B w/ ReasonSeg|44.4|46.0|
> |LISA-7B w/ MMR|49.9|55.6|

---

> > ### Comment · Reviewer_WfTF · 2024-11-24
> >
> > Thanks for the detailed rebuttal. The authors have addressed most of my concerns. Considering the response from the authors and the reviews from other reviewers, I would keep my original rating.

---

### Author Response · Authors · 2024-11-27
**General Response to Reviewers**

We sincerely appreciate the time and effort each reviewer dedicated to providing thoughtful and constructive feedback. We have now submitted a revised version of the paper, incorporating the reviewers’ suggestions.

In summary, we have implemented the following revisions:
- We have included the comparison with GSVA and GLaMM in Table. 2, Table. 3, Table. 4, and Table 5 of Sec 5.2.
- In Appendix A.11, we have added the comparison to validate the comprehensiveness of the MMR dataset against existing reasoning segmentation datasets through zero-shot evaluation
- We have included the performance evaluation on frequently and infrequently occurring object categories and part categories in Appendix A.12.
-  We have included a zero-shot multi-referring segmentation performance comparison on unseen datasets (e.g., GTAV and Cityscapes) in Appendix A.10.
- We have added more examples of the MMR dataset in Appendix A.14.

We are grateful for the reviewers' acknowledgment that all major concerns have been addressed.

A detailed explanation of how each issue was resolved is provided below.

Further discussion is always welcome, and we will do our best to further develop our research.

---

### Meta-Review · Area_Chair_szxa · 2024-12-20

**Metareview:**

# Summary and Recommendation for Acceptance

---

## Strengths:
1. **Innovative Dataset Contribution**:
   - Introduces the **MMR dataset**, a large-scale benchmark with 194K samples, addressing **multi-target** and **multi-granularity reasoning segmentation**.
   - Fills a critical gap by supporting object-level and part-level segmentation in multi-target scenarios, advancing user-interactive vision-language tasks.

2. **Framework and Benchmarks**:
   - Proposes the **M2SA framework**, tailored for multi-target, object-level, and part-level reasoning segmentation tasks.
   - Provides comprehensive benchmarks, demonstrating M2SA’s superior performance over state-of-the-art models (e.g., GSVA, GLaMM).

3. **Thorough Evaluation**:
   - Includes extensive performance comparisons, zero-shot evaluations on unseen datasets (e.g., GTAV, Cityscapes), and analysis of long-tail phenomena.
   - Demonstrates strong generalization across datasets and tasks.

4. **Community Impact**:
   - Addresses an underexplored area in reasoning segmentation with direct applications in robotics and human-AI interaction.
   - Promises public release of the dataset and code, ensuring reproducibility and encouraging further research.

5. **Presentation and Revisions**:
   - The paper is well-written and easy to follow.
   - Authors effectively addressed reviewer concerns by adding additional analyses, comparisons, and dataset examples.

---

## Weaknesses:
1. **Incremental Model Contribution**:
   - The M2SA framework introduces minor modifications (e.g., early local feature fusion, multiple [SEG] tokens), making its novelty incremental.

2. **Auto-annotation Concerns**:
   - Relies on annotations from PACO-LVIS, raising questions about originality. However, the transformation into reasoning-focused question-answer pairs adds significant value.

3. **Performance on Rare Categories**:
   - Slightly lower performance on infrequent object categories, indicating potential room for improvement in handling long-tail distributions.

4. **Initial Comparisons**:
   - Early comparisons with state-of-the-art methods (e.g., GSVA, GLaMM) were limited, though this was resolved during the rebuttal phase.

---

## Authors' Mitigation:
1. **Expanded Comparisons**:
   - Added performance evaluations of GSVA and GLaMM on the MMR dataset, demonstrating the superior performance of M2SA.
   - Conducted zero-shot performance evaluations on unseen datasets (GTAV, Cityscapes) to validate generalization.

2. **Dataset Enhancements**:
   - Addressed auto-annotation concerns by emphasizing the novel reasoning-focused question-answer pairs and the complexity of the queries.

3. **Detailed Analyses**:
   - Analyzed long-tail performance, showing consistency across frequent and infrequent part categories.
   - Provided open-vocabulary segmentation evaluations on new datasets.

4. **Improved Presentation**:
   - Included additional examples and explanations to enhance the clarity and comprehensiveness of the dataset and benchmarks.

---

## Remaining Weaknesses:
1. **Incremental Framework**:
   - The M2SA framework’s contributions remain incremental compared to the dataset’s innovation.
2. **Auto-annotation Limitations**:
   - Despite the mitigations, reliance on PACO-LVIS annotations may restrict perceived originality.

---

## Justification for Acceptance:
The **MMR dataset** addresses a critical gap in reasoning segmentation by enabling research on multi-target and multi-granularity tasks that are pivotal for advanced vision-language systems. While the accompanying M2SA framework lacks significant novelty, its robust performance and the dataset's comprehensive benchmarks underscore the paper's value.

The authors’ detailed rebuttal addressed all reviewer concerns, including expanded comparisons with state-of-the-art methods, in-depth dataset analyses, and additional experiments. The promise of public release of the dataset and code further solidifies its potential impact.

**Recommendation**: Accept.   The dataset's innovation, combined with strong experimental validation and its value to the research community, makes it a worthy contribution to the conference.

**Additional Comments On Reviewer Discussion:**

Please refer to details in the above section.

---

### Decision · Program_Chairs · 2025-01-22

Accept (Poster)